# Many-body quantum dynamics with MCTDH-X

Paolo Molignini[1], Sunayana Dutta[2,3] and Elke Fasshauer[4,5,6]

**1** Department of Physics, Stockholm University, AlbaNova University Center,
106 91 Stockholm, Sweden
**2** Department of Physics, University of Haifa, 3498838 Haifa, Israel
**3** Haifa Research Center for Theoretical Physics and Astrophysics,
University of Haifa, 3498838 Haifa, Israel
**4** Institute of Physical and Theoretical Chemistry, University of Tübingen,
Auf der Morgenstelle 18, 72076 Tübingen, Germany
**5** LISA+, University of Tübingen, Auf der Morgenstelle 15, 72076 Tübingen, Germany
**6** Department of Chemistry, University of Oxford, Oxford OX1 3QZ, United Kingdom

## Abstract

The lecture notes on "Many-body Quantum Dynamics with MCTDH-X", adapted from the 2023 Heidelberg MCTDH Summer School, provide an in-depth exploration of the Multiconfigurational Time-Dependent Hartree approach for indistinguishable particles. They serve as a comprehensive guide for understanding and utilizing the MCTDH-X software for both bosonic and fermionic systems. The tutorial begins with an introduction to the MCTDH-X software, highlighting its capability to handle various quantum systems, including those with internal degrees of freedom and long-range interactions. The theoretical foundation is then laid out on how to solve the time-dependent and time-independent Schrödinger equations for many-body systems. The workflow section provides practical instructions on setting up and executing simulations using MCTDH-X. Detailed benchmarks against exact solutions are presented, showcasing the accuracy and reliability of the software in ground-state and dynamic simulations. The notes then delve into the dynamics of quantum systems, covering relaxation processes, time evolution, and the analysis of propagation for both bosonic and fermionic particles. The discussion includes the interpretation of various physical quantities such as energy, density distributions, and orbital occupations. Advanced features of MCTDH-X are also explored in the last section, including the calculation of correlation functions and the creation of visualizations through video tutorials. The notes conclude with a Linux/UNIX command cheat sheet, facilitating ease of use for users operating the software on different systems. Overall, these lecture notes provide a valuable resource for researchers and students in the field of quantum dynamics, offering both theoretical insights and practical guidance on the use of MCTDH-X for studying complex many-body systems.

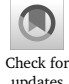
doi:10.21468/SciPostPhysLectNotes.94

# Preamble

This tutorial assumes that you have a working copy of the MCTDH-X software running on your local system or on your supercomputing cluster. MCTDH-X can be run both on a Linux environment and Mac OS. The open-source software MCTDH-X can be downloaded from https://gitlab.com/the-mctdh-x-repository/mctdh-x-releases and instructions on how to install MCTDH-X can be found in the respective folder `documentation/Installation_guide` in the repository. Additional details about the MCTDH-X software are available on the http://ultracold.org website. Details of the MCTDH-X theory, the MCTDH-X code, and the software manual are all located in the 'DOCUMENTATION' section. This tutorial is based mainly on Refs. [1–3]. In this tutorial you will learn how to execute basic calculations with MCTDH-X for both bosons and fermions, extract observables, and plot the results.

# 1 Introduction

The time-dependent many-body Schrödinger equation (TDSE) and its time independent counterpart (TISE) are fundamental equations at the heart of many different fields of science: quantum chemistry, condensed matter physics, and atomic and molecular physics to cite a few. The TDSE describes the dynamics of quantum particles under the influence of their kinetic energy, external potentials, and many-body interactions, whereas the TISE describes the properties of their stationary states. Exact solutions to the TDSE exist only for model systems, like the time-dependent harmonic interaction model [1, 2, 4, 5]. To obtain solutions to the TISE and TDSE, numerical methods as well as their implementation in software are therefore indispensable. One such software is MCTDH-X. MCTDH-X implements the MultiConfigurational Time-Dependent Hartree (MCTDH) approach for indistinguishable quantum particles: bosons [6, 7, 54–57] and fermions [2, 8–14, 57].

In the MCTDH approach, the equations of motions for bosons and fermions can be formulated in a unified manner to enable the description of correlated wavefunctions. This is in stark contrast to mean-field treatments (e.g. Gross-Pitaevskii equation for bosonic systems or Hartree-Fock for fermionic systems) which completely neglect correlations. Furthermore, because the method relies on a time-dependent variational principle, MCTDH-X naturally captures any time dependence present in the parameters of the system.

The current version of MCTDH-X is able to deal with indistinguishable particles with internal degrees of freedom like bosonic spinor condensates [6] (calculations for fermions with spin are currently being developed), indistinguishable particles placed in a high-finesse optical cavity [15–22], indistinguishable particles with short-range [23–31], long-range (e.g. dipolar or regularized Coulomb potentials) interactions [32–36, 36–41, 41–45], quantum particles in complex, often time-dependent potentials [46–51], Hubbard lattice models [52], and has even been interfaced with machine learning [53]. In this tutorial, though, we will restrict to the simplest case of bosons and fermions without additional degrees of freedom. First, we will obtain ground state properties of interacting systems that can be solved exactly. We will systematically compare the numerical convergence of MCTDH-X to the exact solutions. Then, we will use MCTDH-X to simulate and visualize the dynamics of interacting bosons and fermions.

## 1.1 Structure of the MCTDH-X software

### 1.1.1 The theory in a nutshell

The objective of the MCTDH-X software is to numerically solve the TISE or TDSE for a given many-body Hamiltonian and to analyze the computed solutions. This general Hamiltonian describes $N$ interacting, indistinguishable bosons or fermions subject to a one-body potential, and has the form

$$H = \sum_{i=1}^{N} \left[ -\frac{1}{2} \partial_{\mathbf{x}_i}^2 + V(\mathbf{x}_i; t) \right] + \sum_{i<j}^{N} W(\mathbf{x}_i, \mathbf{x}_j; t), \tag{1}$$

that can be either explicitly dependent on time or not. Here, $\mathbf{x}_i$ is the coordinate of the $i$-th particle, $-\frac{1}{2} \partial_{\mathbf{x}}^2$ is the kinetic energy operator, $V(\mathbf{x}; t)$ is a general, possibly time-dependent, one-body potential and $W(\mathbf{x}, \mathbf{x}'; t)$ is a general, possibly time-dependent interparticle interaction operator. All the quantities are given in dimensionless units. In MCTDH-X, the length scale $L$ can be chosen to appropriately represent the physical problem; the corresponding time and energy scales are then determined as $mL^2/\hbar$ and $\hbar^2/(mL^2)$, respectively. For example, in the presence of a harmonic confinement potential, a natural choice for the length scale is the inverse of the harmonic trapping frequency $\omega$, i.e. $L = \sqrt{\hbar/(m\omega)}$. On the other hand, and for

chemical purposes, if the length scale is chosen to be one Bohr radius, the results are given in atomic units.

The TISE corresponding to the Hamiltonian of Eq. (1) is

$$H|\Psi_E\rangle = E|\Psi_E\rangle, \tag{2}$$

while the TDSE is

$$H|\Psi(t)\rangle = i\partial_t|\Psi(t)\rangle. \tag{3}$$

Note that $H$ in the TISE needs to be a time-independent Hamiltonian. In Eq. (2), $|\Psi_E\rangle$ is an eigenstate of $H$ with eigenvalue (energy) $E$. $|\Psi(t)\rangle$ stands for the solution of the TDSE at time $t$.

The MCTDH-X theory [57,58] uses an ansatz for the wavefunction that is a time-dependent superposition of time-dependent many-body basis functions:

$$|\Psi(t)\rangle = \sum_{\vec{n}} C_{\vec{n}}(t)|\vec{n};t\rangle, \quad \vec{n} = (n_1, ..., n_M)^T,$$

$$|\vec{n};t\rangle = \mathcal{N} \prod_{i=1}^{M} \left[\hat{b}_i^\dagger(t)\right]^{n_i} |\text{vac}\rangle, \quad \phi_j(\mathbf{x};t) = \langle\mathbf{x}|\hat{b}_j(t)|0\rangle. \tag{4}$$

Here, the $C_{\vec{n}}(t)$ are referred to as coefficients, the $|\vec{n};t\rangle$ as configurations, and the normalization factor is $\mathcal{N} = \frac{1}{\sqrt{\prod_{i=1}^{M} n_i!}}$ for bosons and $\mathcal{N} = 1$ for fermions. Each configuration is a fully symmetric or fully anti-symmetric many-body basis state built from $M$ orthonormal time-dependent single-particle functions, known as *orbitals*, $\{\phi_k(\mathbf{x},t); k = 1, ..., M\}$. For bosons, the symmetrized configuration state corresponds to a permanent, whereas for fermions the antisymmetrized state is a Slater determinant. To fully specify the solution of the TISE or TDSE, the MCTDH-X software computes and stores the coefficients $C_{\vec{n}}(t)$ and the orbitals $\{\phi_k(\mathbf{x},t); k = 1, ..., M\}$ at times $t$ that are specified by the user. The set of equations of motion for the parameters in Eq. (4) comprises a coupled set of first-order differential equations for time-dependent coefficients $C_{\vec{n}}(t)$ and non-linear integro-differential equations for the orbitals $\phi_j(\mathbf{x};t)$. The MCTDH-X software integrates these equations of motion either in imaginary time (for time-independent Hamiltonians) to obtain the many-body ground-state wavefunction [59], or in real time to perform full-time propagation.

### 1.1.2 Quantities of interest

Once the coefficients $C_{\vec{n}}(t)$ and the orbitals $\phi_k(\mathbf{x},t)$ are computed, the MCTDH-X software can analyze the solution and calculate several quantities of interest. In this tutorial, we will mainly be interested in the total system energy

$$E = \langle\Psi|H|\Psi\rangle, \tag{5}$$

the real-space density distribution

$$\rho(\mathbf{x}) = \langle\Psi|\hat{\Psi}^\dagger(\mathbf{x})\hat{\Psi}(\mathbf{x})|\Psi\rangle/N, \tag{6}$$

and the Glauber one-body and two-body correlation functions

$$g^{(1)}(\mathbf{x},\mathbf{x}') = \frac{\langle\Psi|\hat{\Psi}^\dagger(\mathbf{x})\hat{\Psi}(\mathbf{x}')|\Psi\rangle}{N\sqrt{\rho(\mathbf{x})\rho(\mathbf{x}')}}, \tag{7}$$

$$g^{(2)}(\mathbf{x},\mathbf{x}') = \frac{\langle\Psi|\hat{\Psi}^\dagger(\mathbf{x})\hat{\Psi}^\dagger(\mathbf{x}')\hat{\Psi}(\mathbf{x}')\hat{\Psi}(\mathbf{x})|\Psi\rangle}{N^2\rho(\mathbf{x})\rho(\mathbf{x}')}, \tag{8}$$

where $\hat{\Psi}^{\dagger}(\mathbf{x})$ and $\hat{\Psi}(\mathbf{x})$ are creation and annihilation operators for a particle at position $x$. We will also discuss the occupation of the orbitals for both ground-state and real-time calculations, which give information about numerical convergence and many-body character of the system. These notions are best expressed via the natural orbitals $\phi_i^{(NO)}$ and the orbital occupations $\rho_i$, which are the eigenfunctions and eigenvalues of the reduced one-body density matrix, defined as

$$\rho^{(1)}(\mathbf{x},\mathbf{x}') = \frac{1}{N}\langle\Psi|\hat{\Psi}^{\dagger}(\mathbf{x}')\hat{\Psi}(\mathbf{x})|\Psi\rangle = \sum_i \rho_i \phi_i^{(NO),*}(\mathbf{x}')\phi_i^{(NO)}(\mathbf{x}). \tag{9}$$

In chemistry, these orbitals are called natural atomic orbitals. They are often optimized to maximize occupation numbers, resulting in natural bond orbitals (NBOs). For the latter, the local block eigenfunctions on one atomic center phenomenologically correspond to a lone-pair, while the block eigenfunctions involving two atomic centers correspond to a bond giving rise to the chemist's Lewis structure picture. They are natural in the sense of Löwdin [60], having optimal convergence properties for the description of the electron density.

For bosons, if the eigenvalues are dominated by a single contribution close to unity, the state is well-described by a single coherent single-particle wavefunction. This is often denoted as a Gross-Pitaevksii mean-field description. For fermions, because of the Pauli principle, there must be at least $N$ separate contributions to the eigenvalues. If the contribution from the additional natural orbitals is negligible, the system is well-captured by a mean-field Hartree-Fock description. In our indexing convention, the natural orbitals are ranked in order of decreasing occupation, i.e. $\rho_1 \geq \cdots \geq \rho_M$.

We remark that the capabilities of MCTDH-X extend to the calculation of more quantities not discussed in these lecture notes, such as momentum-space densities and momentum-space correlation functions, single-shot images, full distribution functions, variances, autocorrelation functions, many-body overlaps, many-body entropies, and cavity order parameters [3].

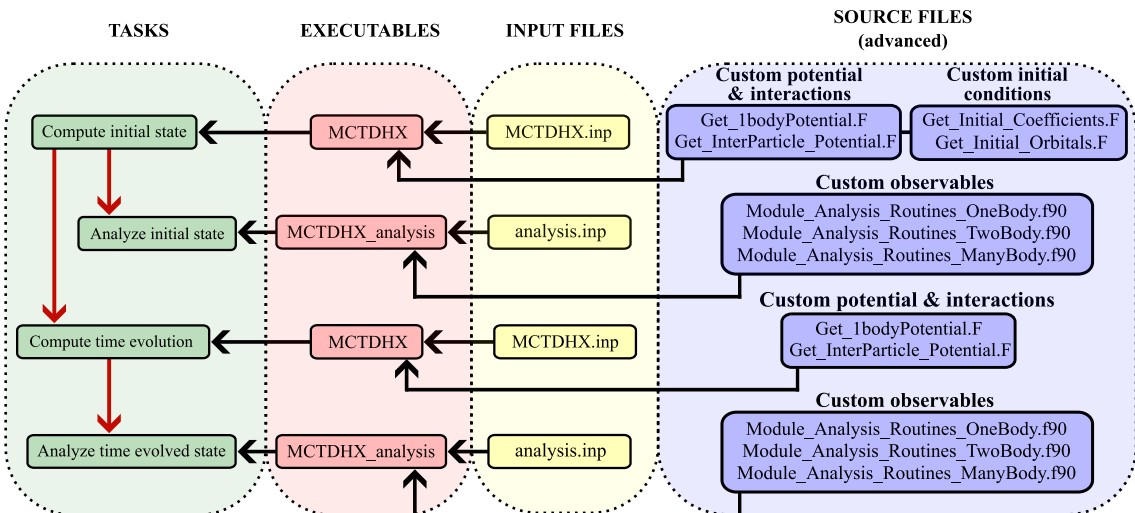

Figure 1: Workflow of the MCTDH-X software. The red arrows indicate the task workflow. The black arrows indicate the setup workflow: first change source files to define custom potential, interactions, initial conditions, and observables (optional), then set up input files, then run executables. In this tutorial, we focus only on the yellow, red, and green parts of the workflow.

## 2 Workflow

The MCTDH-X software consists of two main programs:

1. The main program, `MCTDHX_gcc`, computes the numerical solution of the TISE or TDSE.

2. The analysis program, `MCTDHX_analysis_gcc`, analyzes the found solution to calculate derived observables such as correlations, single-shot images, etc.

To set up a numerical task, the user modifies and chooses the parameters via the text input file `MCTDHX.inp`. If one is interested in the energetics and in the behavior of the real-space density, running the main program is sufficient. A detailed description of the available options is given in the manual [61] of the MCTDH-X software. Here and in the following, we use the `verbatim` font to refer to code, including executable commands, files, statements, and variables.

The workflow and structure of the MCTDH-X software follows naturally from its main objectives to determine a numerical solution to the TISE or the TDSE and then to extract desired quantities of interest from the solution. This workflow can be summarized in the following steps, which are also visualized in Fig. 1.

---

1. **Determine the initial state**. Compute the ground state of some Hamiltonian $\hat{H}$ by running `MCTDHX_gcc` and configuring the numerical task ("relaxation mode", Hamiltonian, integration procedure etc.) in the input file `MCTDHX.inp`.

2. **Analyze the initial state** by choosing the desired quantities of interest in `analysis.inp` and running the analysis program `MCTDHX_analysis_gcc`.

3. **Compute the time-evolution of the initial state** with given Hamiltonian $\hat{H}(t)$. The dynamics of the system is obtained by choosing the numerical task ("propagation mode", Hamiltonian, integration procedure etc.) in the input file `MCTDHX.inp` and running `MCTDHX_gcc`. Typically, the propagation is calculated by restarting from an initial state determined from a relaxation.

4. **Analyze the computed time-evolving state** by choosing the desired quantities of interest in `analysis.inp` and running the analysis program `MCTDHX_analysis_gcc`.

---

## 3 Benchmark against exact solution (ground state)

In this section, we will run our first simulations to obtain the ground-state energy and densities for interacting bosons and fermions for an increasing number of orbitals. We will then see how increasing the number of orbitals leads to progressively more accurate quantitative results. You can run computations for bosons only, for fermions only, or for both. If you have installed the software correctly, you should have already have aliases saved in the `.mctdhxrc` file located in your home directory (`~/`). To activate these aliases, source the file. Open a terminal and type:

```
source ~/.mctdhxrc.
```

Now you will be able to quickly copy the MCTDH-X executables by simply running the command

```
bincp,
```

and to copy sample input files by running

```
inpcp.
```

Another useful alias is

```
cdmx,
```

which will take you directly to the MCDTH-X root directory where the source files, installer etc. are located.

## 3.1 Harmonic interaction model

We will use the harmonic interaction model (HIM) – an exactly solvable many-body problem – to benchmark the validity of our MCTDH-X calculations. The HIM is a model in which quantum particles interact with each other via a harmonic interparticle interaction [1, 2, 62–64]. The first quantized Hamiltonian can be written in dimensionless units as

$$\hat{H} = \sum_{i=1}^{N} \left[ \hat{T}(\vec{r}_i) + \hat{V}(\vec{r}_i) \right] + \sum_{i<j}^{N} \hat{W}(\vec{r}_i, \vec{r}_j), \tag{10}$$

with

$$\hat{T}(\vec{r}) = -\frac{1}{2}\partial_{\vec{r}}^2, \tag{11}$$

$$\hat{V}(\vec{r}) = \frac{1}{2}\omega^2 \vec{r}^2, \tag{12}$$

$$\hat{W}(\vec{r}_i, \vec{r}_j) = K_0(\vec{r}_i - \vec{r}_j)^2, \tag{13}$$

respectively the kinetic energy operator, the one-body potential operator, and the two-body interaction operator. Here, $\omega$ is the frequency of the harmonic trap, and $K_0$ is the strength of the two-body interaction.

The exact solution to this Hamiltonian can be obtained if we employ a coordinate transformation

$$\vec{q}_j = \frac{1}{\sqrt{j(j+1)}} \sum_{i=1}^{j} (\vec{r}_{j+1} - \vec{r}_i), \quad j = 1, \ldots, N-1, \tag{14}$$

$$\vec{q}_N = \frac{1}{\sqrt{N}} \sum_{i=1}^{N} \vec{r}_i, \tag{15}$$

that maps the interacting particles to a collection of non-interacting harmonic oscillators. The transformed Hamiltonian namely reads

$$\hat{H} = \sum_{i=1}^{N-1} \left( -\frac{1}{2}\partial_{\vec{q}_i}^2 + \frac{1}{2}\delta_N^2 \vec{q}_i^2 \right) - \frac{1}{2}\partial_{\vec{q}_N}^2 + \frac{1}{2}\omega^2 \vec{q}_N^2, \tag{16}$$

where $\delta_N = \sqrt{\omega^2 + 2NK_0}$. Note how $\vec{q}_N$ is a center-of-mass operator, i.e. a weighted average over all particle positions, with frequency $\omega$ (fourth term), while the other $N-1$ harmonic

oscillators originate from the set of relative coordinates and all have the same frequency $\delta_N$ (second term).

The separable form of the HIM problem leads to a solution in terms of a product of $N$ generally different harmonic oscillator wave functions. While this approach is general for multiple dimensions, we will for the purpose of this tutorial limit ourselves to the one-dimensional case. The wavefunction then reads

$$\Psi(x_1,...,x_N) = \begin{cases} \left(\dfrac{\delta_N}{\pi}\right)^{\frac{(N-1)}{4}} \left(\dfrac{\omega}{\pi}\right)^{\frac{1}{4}} e^{-\frac{\alpha}{2}\sum_{i=1}^{N} x_i^2 - \beta\sum_{i<j}^{N} x_i x_j}, & \text{for bosons,} \\[2em] \mathcal{N}\left(\dfrac{\delta_N}{\pi}\right)^{\frac{(N^2-1)}{4}} \left(\dfrac{\omega}{\pi}\right)^{\frac{1}{4}} \prod_{1\le i<j\le N}(x_i - x_j) e^{-\frac{\alpha}{2}\sum_{i=1}^{N} x_i^2 - \beta\sum_{i<j}^{N} x_i x_j}, & \text{for fermions,} \end{cases}$$

(17)

where $\prod_{1\le i<j\le N}(x_i - x_j) = V(x_1,...,x_N)$ is the Vandermonde determinant. The relative Jacobi coordinates of fermions give rise to a dimensionless normalization constant, $\mathcal{N}$ in the wavefunction [65,66]. The coefficients $\alpha$ and $\beta$ in the wavefunction are written as [66],

$$\alpha = \delta_N\left[1 + \frac{1}{N}\left(\frac{\omega}{\delta_N} - 1\right)\right], \qquad \beta = \alpha - \delta_N.$$

The corresponding total ground-state energy is then simply the sum of the ground-state energy of each oscillator.[1] For bosons, this gives

$$E_{\text{exact,bosons}} = E_{\text{rel}} + E_{\text{c.o.m.}} = \left(\frac{N-1}{2}\delta_N + \frac{\omega}{2}\right).$$

(18)

For spin-polarized fermions, the energy is slightly different on account of their different statistics[2] that leads us to sum $N-1$ harmonic oscillator energies $\epsilon_n = (n + 1/2)\delta_N = \frac{2n+1}{2}\delta_N$, yielding

$$E_{\text{exact,fermions}} = \left[\sum_{n=1}^{N-1}\left(\frac{2n+1}{2}\delta_N\right) + \frac{\omega}{2}\right] = \left(\frac{N^2-1}{2}\delta_N + \frac{\omega}{2}\right).$$

(19)

## 3.2 Simulations

We can now use the HIM above to benchmark the validity of our MCTDH-X calculations. The first thing to do when preparing a new computation is to create a separate folder where the program will run. We will treat bosonic and fermionic computations separately, as spin-polarized fermions always require at least $M > N$ orbitals due to the Pauli exclusion principle, whereas for bosons we can perform calculations with $M < N$ since multiple bosons can occupy the same state.

For bosons, we will consider $N = 10$ particles and run 5 different computations with an increasing number of orbitals from 1 to 5. To that end, we need to create 5 different folders. A good structure could be the one shown in Fig. 2, with a parent folder called HIM-bosons and five subfolders

For fermions, we will run several computations with increasing particle number and orbitals:

- Four computations with $N = 2$ particles and $M = 3, 4, 5, 6$ orbitals.

- Three computations with $N = 5$ particles and $M = 6, 7, 8$ orbitals.

---

[1]Recall that the ground-state energy of a quantum harmonic oscillator of frequency $\omega$ is $E_0 = \frac{\hbar\omega}{2}$. In this case, since we operate in dimensionless units, we have effectively set $\hbar = 1$.

[2]Fermions will occupy one by one all the equally-spaced harmonic oscillator levels due to the Pauli principle.

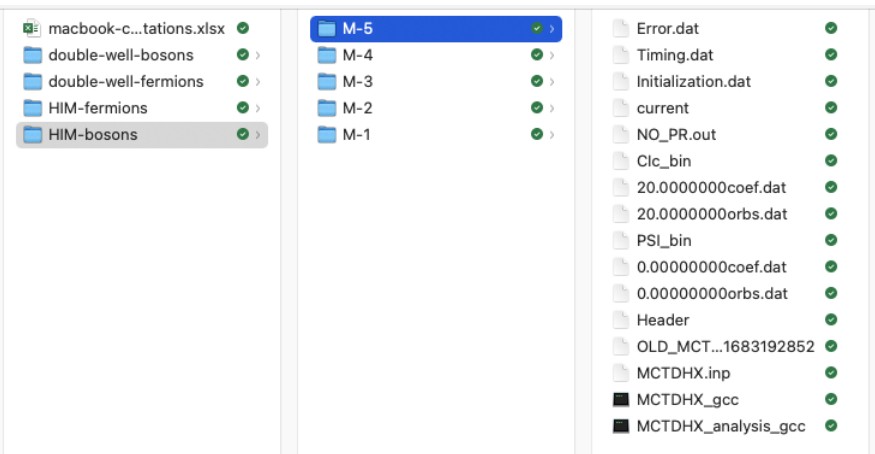

Figure 2: Folder structure for the ground-state bosonic computations.

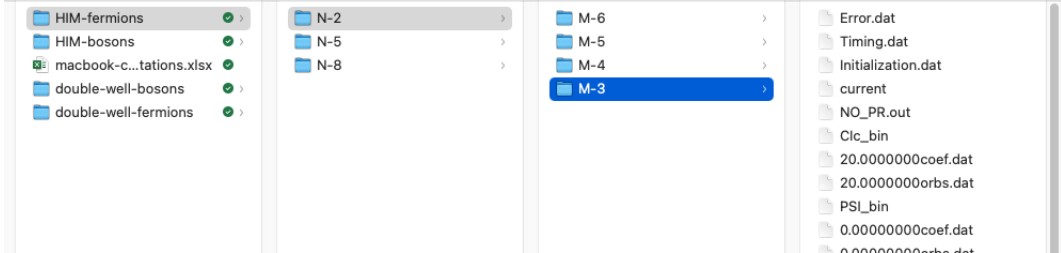

Figure 3: Folder structure for the ground-state fermionic computations.

- Two computations with $N = 8$ particles and $M = 9, 10$ orbitals.

Because we will run simulations of many different cases, it would be best to create separate folder for each case. A good structure could be the one shown in Fig. 3, with a parent folder called `HIM-fermions` containing folders labelled by $N$, which in turn contain folders labelled by $M$.

Next, we need to import the MCTDH-X executables and input files. You already know how to do it, just navigate to each folder and type `bincp` and `inpcp` in your console. You should now have the following files: `MCTDHX.inp`, `analysis.inp`, `MCTDHX_analysis_gcc`, `MCTDHX_gcc`. Since we are currently only focusing on performing a relaxation to the ground state to calculate its energy, we can ignore the analysis files for now. To launch our simulation, we first need to modify the input file. Go ahead and use your favorite text editor to open the file `MCTDHX.inp` (e.g. `vim`). The input file should look like the screenshot of Fig. 4.

To set up your computation, you need to change several parameters in the input file. In the following list, we only mention the parameters that need to be changed to set up the current relaxation for harmonic interacting bosons and fermions. As you can see from looking at `MCTDHX.inp` there are many more parameters that can be modified. They are grouped by their function. For instance, there is an entire section of the input file dedicated to parameters used to model multilevel systems (fermions/bosons with internal degrees of freedom), another for Bose-Hubbard lattice Hamiltonians, another for BEC-cavity systems etc. We will not discuss the functionalities of these parameters in this tutorial. Some other parameters are for advanced control of the simulation in terms of how to discretize the coordinate space, which integrator to use, how to handle errors, etc. Unless explicitly stated, you should not modify any other parameter that does not appear on the list below.

- `JOB_TYPE`: this flag tells the program whether it should simulate bosons, fermions, or

use the full configuration interaction method (debugging purposes only). For **bosonic** computations, you should set JOB_TYPE = 'BOS'.

For **fermionic** computations, you should set JOB_TYPE = 'FER'.

Note that lines that start with an exclamation mark (!) are comment lines and will not be read by the program.

- Morb: this parameter controls the number of orbitals used in the simulation. For **bosons**: set it to the number of orbitals for your computation (1 to 5).

  For **fermions**: set it to the number of orbitals for your computation (3 to 10, depending on the value of $N$).

  Note that for spin-polarized fermions obeying Pauli exclusion, we require *at least $M = N$* orbitals. However, this case can be degenerate in MCTDH-X, so make sure to always choose $M \geq N + 1$.

- Npar: this parameter controls the number of particles used in the simulation. Since we are benchmarking our results against the ones available in Refs. [1] and [2], we should use the same number of particles.

  For **bosons**: set Npar = 10.

  For **fermions**: set Npar = 2, Npar = 5, or Npar = 8.

- xlambda_0: this parameter corresponds to the interparticle interaction strength ($K_0$ in the notation of the previous section).

  For **bosons**: in Ref. [67], $K_0(N - 1) = 0.5$, or $K_0 = 0.05\bar{5}$. Thus set xlambda_0 = 0.05555555556.

  For **fermions**: In Ref. [2], $K_0 = 0.5$. Thus set xlambda_0 = 0.5.

- mass: this parameter controls the mass of the particles. In dimensionless units, mass = 1.0d0 by default.

- Job_Prefactor: this is an ordered pair of numbers (equivalent to a complex number) that determines whether the simulation will be conducted in imaginary time (relaxation) or real time (propagation). Since we are interested in doing a relaxation to the ground state now, you should set it to Job_Prefactor = (-1.0d0,0.0d0).

- GUESS: this flag specifies the initial condition for the simulation. GUESS = 'HAND' is the default for relaxations and makes the integrator start from a random initial state (or

```
  1 System_Parameters
  2 !c==========
  3 !c========== System definition
  4 !c==========
  5 JOB_TYPE = 'BOS'                  !c========== Standard MCTDHB
  6 !JOB_TYPE = 'FER'                 !c========== Standard MCTDHF
  7 !JOB_TYPE = 'FCI'                 !c========== Full CI
  8
  9 Morb = 1                         !c========== Number of Orbitals
 10 Npar = 10                        !c========== Number of Particles
 11 xlambda_0 = 0.05555555556e0              !c========== Strength of the interparticle interaction
 12 mass = 1.0d0
 13 Job_Prefactor = (-1.0d0,0.0d0)   !c========== (0.0d0,-1.0d0) Forward propagation
 14                                  !c========== (0.0d0,+1.0d0) Backward propagation
 15                                  !c========== (-1.0d0,0.0d0) "Improved" Relaxation
 16 !GUESS = 'BINR'                  !c========== Restart from binary files
 17 !GUESS = 'DATA'                  !c========== Restart from ASCII files
 18 GUESS = 'HAND'                   !c========== Start from the initial state specified in the Get_Initial_Orbitals and Get_Initial_Coefficients routines.
 19
 20 NProjections = 2                 !c========== How often to apply the projection operator
 21 Diagonalize_OneBodyh = .F.       !c========== Start from the eigenfunctions of the one-body Hamiltonian h as orbitals.
 22 Binary_Start_Time = 0.0d0        !c========== If GUESS='BINR' start from this point in time.
 23 Restart_State = 1                !c========== State from which to restart from a previous block relaxation.
 24 Restart_Orbital_FileName = '10.0000000orbs.dat'      ! If GUESS='DATA' then the orbitals are read from this ASCII file.
 25 Restart_Coefficients_FileName = '10.0000000coef.dat' ! If GUESS='DATA' then the coefficients are read from this ASCII file.
 26 Custom_Orbital_Initialization = .F. !c========== If GUESS='HAND' Do we use custom initial orbital?
 27 Which_Custom_Orbital_Initialization = 'Random' !c========== If GUESS='HAND' Which custom initial orbital do we use?
 28 Custom_Orbital_Initial_Parameters = 0.d0,0.d0,0.d0,0.d0,0.d0,0.d0,0.d0,0.d0 !c========== Parameters for specifying the initial orbital.
 29
 30 Fixed_LZ = .F.                   !c========== In relaxations: Apply a projection to a fixed orbital phase profile?
 31 OrbLz = 0,0,0,0,0,0,0,0,0,0      !c========== How many times 2\pi the phase is going to jump. If any value is -666, the respective orbital is unchanged.
 32 Vortex_Seeding = .F.             !c========== In propagations: Multiply initial orbitals with phase/density profile?
 33 Vortex_Imprint = .F.             !c========== In relaxations: Apply a projection operator to a certain orbital density profile?
 34 Profile = 'tanh'                 !c========== If Vortex_Imprint is true, this will select the respective shape of the imprinted profile.
 35 NonInteracting = .F.             !c========== Omit two-body part of the Hamiltonian?
 36 !c==========
```

Figure 4: First lines in the input file MCTDHX.inp.

a custom initial state if provided). Later on when we perform time evolution, we will use `GUESS = 'BINR'` to restart a computation from binary files belonging to a converged relaxation.

- `DIM_MCTDH`: this parameter determines the dimensionality of the problem. For our benchmark, we will focus on one-dimensional systems and set `DIM_MCTDH=1`.

- `NDVR_X, NDVR_Y, NDVR_Z`: These three parameters set the number of points in the x, y, and z directions in the Discrete Variable Representation (DVR) of the continuum system.[3] Since we are only interested in a 1D problem for now, we can set `NDVR_Y=1` and `NDVR_Z=1`. It is always best to set the number of DVR points to be a power of 2 to avoid numerical rounding errors in binary representation. The number of points should be chosen to be compatible with the *size* of the spatial grid (see next point), i.e. the larger the grid, the more points we require to keep the same accuracy. Typically, `NDVR_X=128` is enough to obtain good results.

- `x_initial, x_final, y_initial, y_final, z_initial, z_final`: this set of 6 parameters control the spatial extension of the DVR grid, for all three possible directions. For this simulation, you can set `x_initial = -8.0d0` and `x_final = +8.0d0`. Because we are dealing with a 1D problem, the extent of the other spatial dimensions will be ignored.

- `Time_Begin`: This parameter determines the starting time of the computation. Standard is `Time_Begin = 0.0d0`. A different time is only chosen if you want to continue the propagation from a terminated calculation from the last calculated point using the stored orbitals and coefficients.

- `Time_Final`: This parameter determines the final time of the computation, in dimensionless units (i.e. the program will integrate the equations of motion until this time). Heuristically, `Time_Final = 20.0d0` is typically enough to achieve converged results in the energy, i.e. results whose energy does not change in the first 12 decimal digits even when letting the system time evolve further. Note that this does *not* necessarily mean that the numerical values of the calculated energies are exact! This only means that with the current number of orbitals, Krylov space, type of integrator etc. that's the best precision possible. As you will see later, increasing the number of orbitals lead to progressively better (=lower) ground state energies.

- `Output_TimeStep`: this flag tells the program when to write output data in the form of `.orbs` and `.coef` files. If you would like to see these files generated throughout the imaginary time evolution, set something like `Output_TimeStep = 1.0d0`. Otherwise, `Output_TimeStep = 20.0d0` will only produce an initial output (at time 0) and a final output (at time 20).

- `Integration_Stepsize`: this parameter indicates the step size used in the integrator. The standard value for a relaxation is `Integration_Stepsize = 0.1d0`. The standard value for a real time propagation is `Integration_Stepsize = 0.01d0` (because it's more prone to accumulation errors).

- `Write_ASCII`: this flag should be set to true, i.e. `Write_ASCII = .T.` if `.orbs` and `.coef` ASCII files ought to be generated.

---

[3]If you want to know more about discretization methods, you can take a look at these lecture notes.

- `Coefficients_Integrator`: MCTDH-X has many different integrators implemented, both for the orbital equation of motion (not listed here) and for the coefficients. While typically you can always use a Runge-Kutta integrator for the orbitals (`Orbital_Integrator = 'RK'`), the coefficient integrator differs depending on whether you are performing a relaxation or a propagation. For a relaxation, set `Coefficients_Integrator ='DAV'` (Davidson diagonalization). For a propagation, set `Coefficients_Integrator ='MCS'` (MCTDH short iterative Lanczos routine).

- `whichpot`: this parameter determines the type of one-body potential used in the simulation. There are many potentials listed in the manual and coded in the module `Get_1bodyPotential.F`. In this tutorial, we are interested in a one-dimensional harmonic oscillator, which corresponds to setting `whichpot="HO1D"`.
  The more experienced user will later be able to add customized one-body potentials to the above mentioned list.

- `parameter1` to `parameter30`: each one-body potential comes with its own set of parameters that can be modified to control its shape. There can be presently up to 30 parameters to specify the one-body potential. For a simple one-dimensional harmonic oscillator centered around the origin and codified by HO1D, there is only one nonzero parameter corresponding to the trapping frequency: `parameter1=1.d0`.

- `Interaction_Type`: this technical flag controls the type of interparticle interaction and how it should be evaluated in the equations of motion. The sample input file provides a brief explanation of all the different types. For time-independent long-ranged interactions, type 4 or type 7 should be used. For harmonic interactions, set `Interaction_Type=4`.

- `which_interaction`: this parameter selects the physical type of interparticle interaction to be simulated (if it is not contact interactions). Each interaction is codified in a 5-6 letter string (see the manual for more information). For harmonic interactions, you should set `which_interaction='HIM'`.

After having set up the input file, you are now ready to launch your first MCTDH-X computation! In the folder of choice (corresponding to $M = 1$, $M = 2$ etc.) where the input file is located, simply run

```
./MCTDHX_gcc.
```

You will see output on the terminal. In particular, you can see the convergence of the energy as a function of the integration time step.

## 3.3 Discussion of the results

After you have run MCTDH-X relaxations in all the folders with different orbital numbers, we can compare the results and see how the numerical solution converges to the exact one as we increase $M$. You will see that several different files are generated. In particular:

- `PSI_bin` and `CIc_bin` are the binary representations of orbitals and coefficients, respectively.

- `.orbs` files contain information about the orbitals at every (imaginary) time step that you have chosen to output.

- `.coef` files contain information about the coefficients at every (imaginary) time step that you have chosen to output (if $M > 1$).

- `NO_PR.out` contains the progression of the energy, and the orbital occupations as a function of (imaginary) time.

- `Header` is a header file that contains information about which parameters were used in the computation and is needed to restart computations from a previous time, or to run propagations from relaxations.

- `Initialization.dat` contains log information about the initialization.

- `Error.dat` contains log information about integration errors, matrix inversion errors etc.

- `Timing.dat` contains log information about how long it took for each step in the program execution to run.

Let us explore the physics of the problem by using the files that MCTDH-X produced.

### 3.3.1 Energy

The energy of a relaxation can easily be read from the standard output while the computation is running. It can also be read from the `NO_PR.out` file. The energy as a function of time is contained in the $(M+2)$-th column. To get the full time dependence of the energy, you can use:

```
awk '{ print $NF }' NO_PR.out.
```

If you are only interested in energy of the very last time step, you can write:

```
awk '{ print $NF }' NO_PR.out | tail -n1.
```

We now discuss the results for bosons and fermions separately.

**Bosons**    In table 1, you can see a comparison of the ground-state bosonic energies as a function of number of orbitals and the times needed to compute them on a 2 GHz Quad-Core Intel Core i5. You can also see that as we increase the number of orbitals, the numerical solution converges to the exact one exponentially. In Fig. 5 we visualize the relationship between the computation time and the energy convergence displayed in table 1. Locally, we are restricted to $M = 5$ orbitals if we want to keep the computation within a short amount of time. Already for $M = 5$ the computation takes around 25 minutes. This is because the size of the configuration space scales roughly as $\begin{pmatrix} N + M - 1 \\ N \end{pmatrix} \approx N^M$ when $N \gg M$.[4] Note, however, that in higher dimensions and on a cluster (actually on any multi-core system), MCTDH-X can use a decomposition of the orbitals in smaller domains that allows for parallelization and faster computation times.

We can also plot the energy visually, for instance using the open-source software `gnuplot`. The syntax for $M = 5$ is for instance

```
plot "NO_PR.out" u 1:7 w l lw 3.
```

Plots for the energy convergence as a function of imaginary time for increasing $M$ are shown in Fig. 6. Try to recreate this plot with `gnuplot` or another visualization software.

---

[4]This approximation is *very* crude. In fact, the scaling is superexponential because of the factorials and since we typically use finite/small values for $N$ and $M$, the Stirling approximation is not very accurate here.

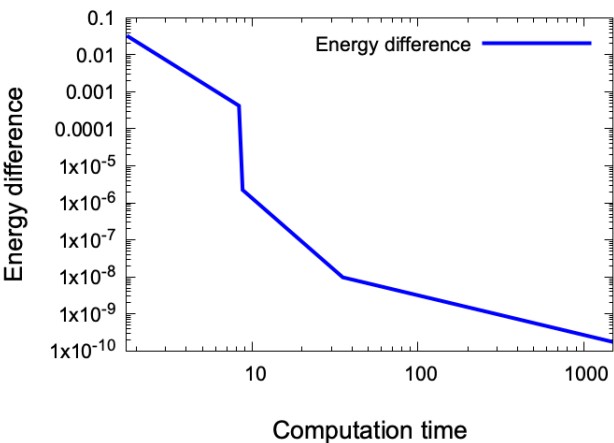

Figure 5: Relationship between the computation time and the energy convergence for $N = 10$ harmonically-interacting bosons and increasing orbital number $M$, as shown in table 1.

Table 1: Approximate computation times and the corresponding ground-state energies after 20 time steps for $N = 10$ harmonically-interacting bosons and increasing orbital number $M$. The calculations were performed on a 2 GHz Quad-Core Intel Core i5, 16 GB 3733 MHz LPDDR4X RAM. The bold numbers correspond to the digits that are equal to the exact solution.

| Orbital number $M$ | Approximate computation time | Energy at time step 20 |
|---|---|---|
| 1 | 1.75 seconds | **7.07**1067812008208 |
| 2 | 8.31 seconds | **7.038**769026440956 |
| 3 | 8.74 seconds | **7.0383**50652543779 |
| 4 | 35.19 seconds | **7.0383484**25047187 |
| 5 | 25 minutes | **7.038348415**486683 |
| exact solution | | 7.038348415311011 |

**Fermions** In table 2, you can see a comparison of the fermionic ground-state energies as a function of number of orbitals and the times needed to compute them on a Quad-Core Intel Core i5. You can also see (at least for $N = 2$ where we have more data points) that as we increase the number of orbitals for a fixed particle number, the numerical solution converges to the exact one exponentially. On a local computer, we are restricted to around $N = 8$ particles and $M = 10$ orbitals if we want to keep the computation within a short amount of time (in the order of minutes). Much like for bosons, the size of the configuration space for fermions scales exponentially, roughly as $\begin{pmatrix} M \\ N \end{pmatrix} \approx \frac{M^N}{N!}$.

Plots for the energy convergence as a function of imaginary time for $N = 2$ and increasing $M$ is shown in Fig. 7. Try to recreate this plot with `gnuplot` or another visualization software. To visualize the energy convergence with `gnuplot`, for instance for $N = 2$, $M = 5$, the syntax is

```
plot "NO_PR.out" u 1:7 w l lw 3.
```

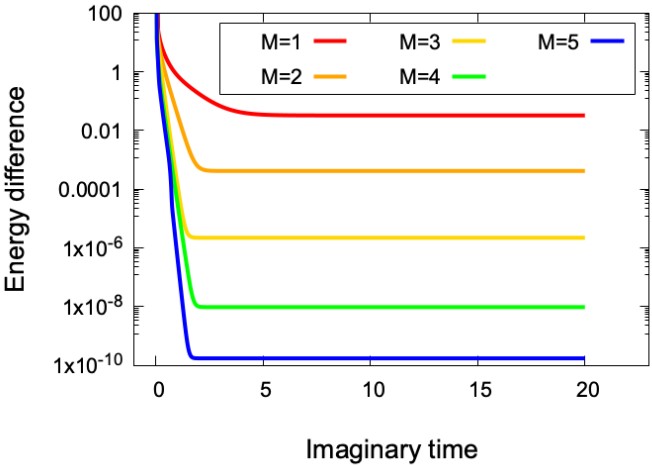

Figure 6: Imaginary-time convergence of the ground-state energy – calculated as $E_M(it = 20) - E_{\text{exact}}$ – for $N = 10$ harmonically interacting bosons in a harmonic potential as a function of increasing orbital number $M$.

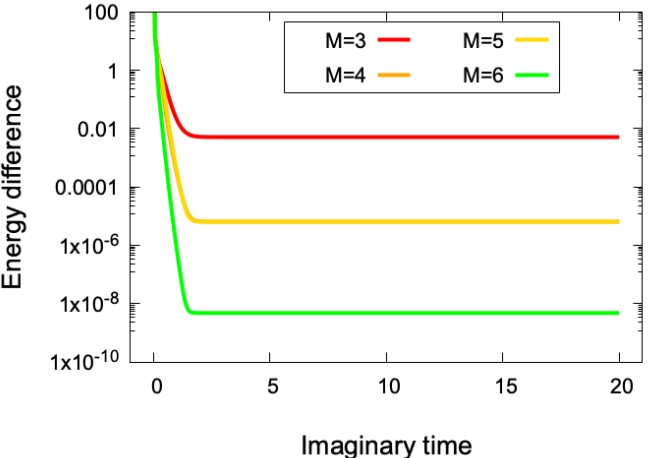

Figure 7: Imaginary-time convergence of the ground-state energy – calculated as $E_M(it = 20) - E_{\text{exact}}$ – for $N = 2$ harmonically interacting fermions in a harmonic potential as a function of increasing orbital number $M$.

### 3.3.2 Orbital occupation

We have seen that a higher number of orbitals contributes to a better result for the ground-state energy. It might also be of interest to quantify and visualize this trend in another way. This can be done by plotting the orbital occupations, which are saved in the $M$ columns of the NO_PR.out file before the energy column (i.e. column 2 to $M + 1$). This is an important procedure because it tells us whether our results are converging in the chosen number of orbitals (i.e. adding more orbitals does not change the many-body wavefunction by much) or not.

In the case of bosons, this procedure can help us quantify how much correlation we need to include in our calculations to observe the correct physics, think for instance of a superfluid vs. a Mott insulating regime. In the case of fermions, a nonzero occupation in more than $N$ orbitals quantifies how much the many-body wave function deviates from a single Slater determinant and thus how much quantum entanglement is building up in the system [68].

Table 2: Approximate computation times on a 2 GHz Quad-Core Intel Core i5, 16 GB 3733 MHz LPDDR4X RAM and the corresponding ground-state energies after 20 time steps. The bold numbers correspond to the digits that are equal to the exact solution.

| Particle number | Orbital number $M$ | Approximate computation time | Energy at time step 20 |
|---|---|---|---|
| 2 | 3 | 7.75 seconds | **3.**103244922155683 |
| 2 | 4 | 14.11 seconds | **3.098**082754434019 |
| 2 | 5 | 25.32 seconds | **3.098**082754434124 |
| 2 | 6 | 50.24 seconds | **3.09807621**6222958 |
| exact solution | | | 3.0980762113533159 |
| 5 | 6 | 42.95 seconds | **29.9**3368010792828 |
| 5 | 7 | 72.04 seconds | **29.89**850111772063 |
| 5 | 8 | 112.40 seconds | **29.89**443423675234 |
| exact solution | | | 29.893876913398137 |
| 8 | 9 | 166.86 seconds | **95.08**244224010051 |
| 8 | 10 | 247.87 seconds | **95.0**1531056595735 |
| exact solution | | | 95.000000000000000 |

Another way to phrase this is that the orbital occupation tells us whether our results can be described by mean-field theories like the Gross-Pitaevskii equation for bosons (equivalent to using a single orbital, i.e. setting $M = 1$) or the Hartree-Fock approximation for fermions (equivalent to using $M = N$), or whether higher orbitals are important. This is also very important for real-time dynamics, because correlated states might appear as a function of time.

Let us plot the orbital occupation for our relaxations, again considering bosonic and fermionic results separately.

**Bosons**  To plot the bosonic orbital occupations with gnuplot for the $M = 5$ case, write

```
plot "NO\_PR.out" u 1:2 w l lw 3 title "fifth orbital",
"NO\_PR.out" u 1:3 w l lw 3 title "fourth orbital",
"NO\_PR.out" u 1:4 w l lw 3 title "third orbital",
"NO\_PR.out" u 1:5 w l lw 3 title "second orbital",
"NO\_PR.out" u 1:6 w l lw 3 title "first orbital".
```

Fig. 8 shows precisely this. You can see that the convergence in imaginary time to the final orbital configuration is quite fast. Furthermore, the problem is mainly dominated by one orbital, so a mean-field description is enough to describe the qualitative behavior of the system. However, if we want to come close to the *quantitative* value of the ground-state energy, we must include higher orbitals and in fact their population is nonzero. The inset in Fig. 8 shows an enlarged plot around zero, and we can clearly see that the second orbital is contributing to about 0.3%, while the contributions of higher orbitals are even smaller.

The orbital occupations at the end of the imaginary time evolution can be extracted with

```
awk '{ print $2, $3, $4, $5, $6 }' NO_PR.out | tail -n1,
```

and are reported in table 3.

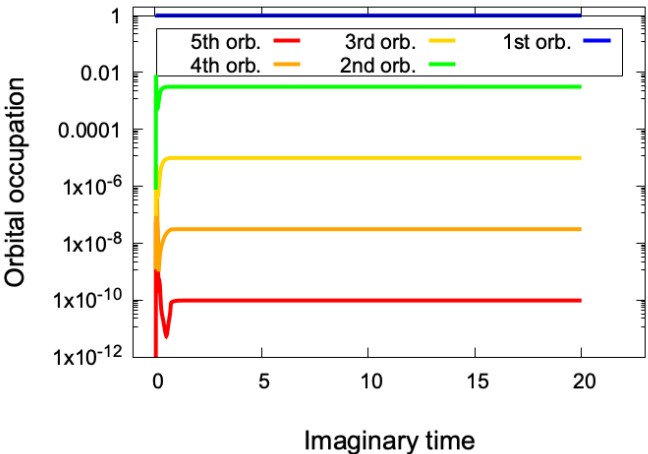

Figure 8: Orbital occupation in the ground state for $N = 10$ harmonically-interacting bosons in a harmonic trap described by $M = 5$ orbitals. Note that in the numerics the orbital occupations are normalized such that they sum to 1 and not $N$.

Table 3: Orbital occupation in the ground state for $N = 10$ harmonically-interacting bosons in a harmonic trap described by $M = 5$ orbitals. The computations were performed on a 2 GHz Quad-Core Intel Core i5, 16 GB 3733 MHz LPDDR4X RAM.

| Orbital | Occupation |
|---------|------------|
| 1st | 0.9968427274690541 |
| 2nd | 0.3147304166934598E-02 |
| 3rd | 0.9936893501857101E-05 |
| 4th | 0.3137224570732360E-07 |
| 5th | 0.9826348708153463E-10 |

**Fermions**    The visualization of the orbital occupation for fermions is similar to what we have already done for bosons. To plot the orbital occupations with gnuplot for e.g. the $N = 2$, $M = 5$ case, write

```
plot "NO\_PR.out" u 1:2 w l lw 3 title "fifth orbital",
"NO\_PR.out" u 1:3 w l lw 3 title "fourth orbital",
"NO\_PR.out" u 1:4 w l lw 3 title "third orbital",
"NO\_PR.out" u 1:5 w l lw 3 title "second orbital",
"NO\_PR.out" u 1:6 w l lw 3 title "first orbital".
```

This is shown in Fig. 9. Once again, the imaginary time convergence to the final orbital configuration is very fast. In this case, the system is very well described by just two orbitals that account for around 50% of the overall population each. In the inset of Fig. 9 we additionally show a zoom of the occupation around zero, where we can distinguish an additional but very small occupation of the third orbital of about 0.04%. These results indicate that the many-body wavefunction is very well described by the usual Slater determinant construction and there is not a lot of correlation between the two fermions.

The orbital occupations at the end of the imaginary time evolution can be extracted with

```
awk '{ print $2, $3, $4, $5, $6 }' NO_PR.out | tail -n1,
```

and are listed in table 4.

Table 4: Orbital occupation in the ground state for $N = 2$ harmonically-interacting fermions in a harmonic trap, described by $M = 5$ orbitals. The computations were performed on a 2 GHz Quad-Core Intel Core i5, 16 GB 3733 MHz LPDDR4X RAM.

| Orbital | Occupation |
|---------|------------|
| 1st | 0.4995067585437437 |
| 2nd | 0.4995067585437435 |
| 3rd | 0.4932414562564047E-03 |
| 4th | 0.4932414562563642E-03 |
| 5th | 0.1493748698862856E-17 |

### 3.3.3 Density

Finally, let us take a look at the density of the many-body wavefunction. This is saved in the sixth column of the `.orbs` files for every time step that you chose to output. If we want to check the shape of the density in the ground state, we simply plot this data for the last time step in the relaxation with

```
plot "20.0000000orbs.dat" u 1:6 w l lw 3.
```

**Bosons** The plot of the density is shown in Fig. 10, together with a comparison with noninteracting bosons in a harmonic trap. You can easily generate the latter results by performing another simulation and choosing $\lambda_0 = 0.0$. As we can see (and not surprisingly), the (attractive) harmonic interaction makes the particles clump together more at the bottom of the trap.

**Fermions** The plot of the density for $N = 5$ fermions is shown in Fig. 11, together with a comparison with the same number of noninteracting fermions in a harmonic trap. As for the bosons, the attractive harmonic interaction makes the fermions clump together more at the bottom of the trap and reduces the typical ripple-like pattern caused by the Pauli exclusion principle of noninteracting fermions in a harmonic trap. Nevertheless, when compared against

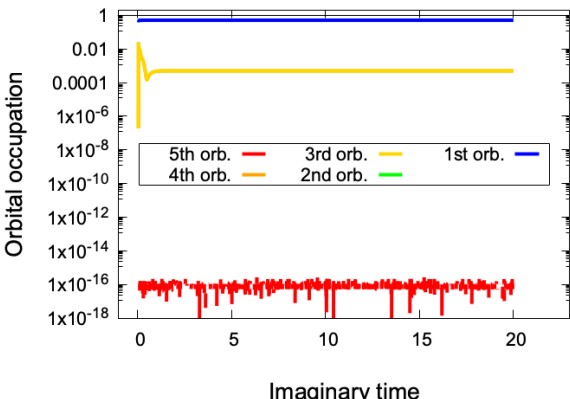

Figure 9: Orbital occupation in the ground state for $N = 2$ harmonically-interacting fermions in a harmonic trap, described by $M = 5$ orbitals. The occupations of the first and second, and of the third and fourth orbital are degenerate. Note that in the numeric the orbital occupations are normalized such that they sum to 1 and not $N$.

bosons, we still see that fermions occupy a larger space due to the repulsion stemming from the Pauli principle.

### 3.3.4 Additional exercises

- Try to plot the density for all the computations with different $M$ in the same plot. Can you see a difference? How does this result relate to the orbital occupation results?

- Try to recreate the same results, but for *repulsively* harmonically interacting bosons and fermions (you need to choose a negative $K_0$). Be aware that if you choose a too large value of $|K_0|$, convergence issues will arise because for harmonic repulsions, the further away the particles are from each other, the stronger they will repel. How do the results look like in terms of energy, occupation, and density? How do they compare to the attractive or noninteracting case?

## 4 Dynamics

So far we have effectively solved the TISE to obtain information about the many-body ground state. Since MCTDH-X optimizes both orbitals and coefficients through the time-dependent variational principle, though, it is also capable of solving the TDSE to capture the dynamics of a system described by a time-dependent Hamiltonian. The time dependence can be of any sort. It can stem from a periodic drive, it can describe sudden quenches of parameters, or it can mirror a full experimental protocol with any kind of ad-hoc time dependence.

To explore this functionality, here we discuss systems of bosons or fermions in a harmonic trap with a time-dependent barrier. We first use MCTDH-X to relax the system into its ground state. Then, we linearly ramp up a central barrier and study the response of the quantum particles.

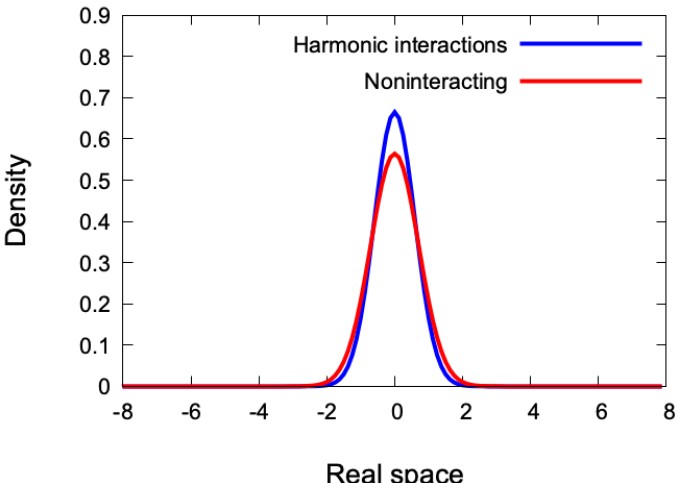

Figure 10: Many-body density for $N = 10$ bosons and $M = 5$ orbitals in a harmonic trap. The blue curve corresponds to harmonically interacting bosons, while the red curve shows noninteracting bosons.

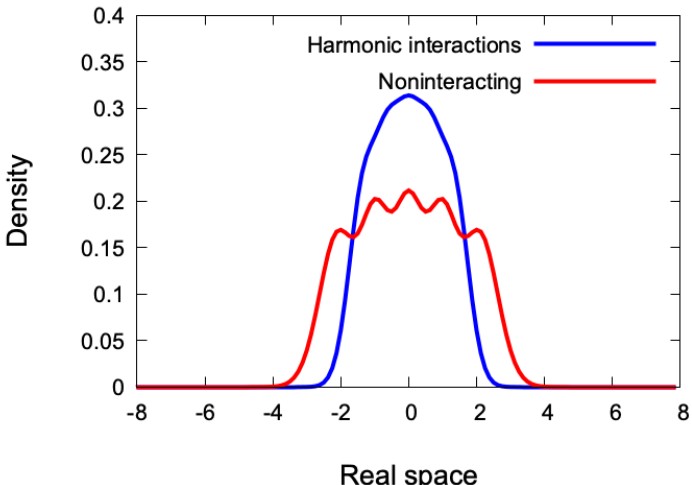

Figure 11: Many-body density for $N = 5$ fermions and $M = 8$ orbitals in a harmonic trap. The blue curve corresponds to harmonically interacting fermions, while the red curve shows noninteracting fermions.

## 4.1 Relaxation

We will explore the time evolution of a system of quantum particles confined in a harmonic trap with a time-dependent Gaussian barrier, creating an effective double well.

The two-body interaction between the **bosons** is chosen as a contact repulsion of the form

$$\hat{W}(\vec{x}_i, \vec{x}_j) = W_0 \delta(x_i - x_j),\tag{20}$$

where $W_0 = 1$ is the interaction strength and $\delta$ is the Dirac delta distribution.

For **fermions**, we will consider interactions that do not have an analytical solution, namely regularized Coulomb repulsion:

$$\hat{W}(\vec{r}_i, \vec{r}_j) = \frac{W_0}{\sqrt{|\vec{r}_i - \vec{r}_j|^2 + \alpha \exp(-\beta |\vec{r}_i - \vec{r}_j|)}},\tag{21}$$

with $W_0 = 2.0$, $\alpha = 0.01$, $\beta = 100$. This interaction type is labelled as `regC` in the module `source/ini_guess_pot/Get_InterParticle_Potential.F`. You can have a look at its implementation in this module if you like.

To evaluate the dynamics, first we need to prepare the system in a ground state. Hence, we will relax $N = 5$ bosons or fermions by employing MCTDH-X in a 1D harmonic trap (with imaginary time propagation) and then evolve this initial state with a time-dependent Hamiltonian (with real time propagation).

For the ground state simulation, create a new folder (e.g. named relax). Then, type `inpcp` to load the input files and `bincp` to load the executable files. Now perform the relaxation by modifying `MCTDHX.inp` file as in the following:

- `JOB_TYPE='BOS'` for bosons or `JOB_TYPE='FER'` for fermions,

- `Npar=5`,

- `Morb=10`,

- `xlambda_0=1.0d0` (this parameter corresponds to interaction strength, $W_0$),

- `mass=1.0d0`,

- `Job_Prefactor=(-1.0d0,0.0d0)` (for relaxation),

- `GUESS='HAND'`,

- `DIM_MCTDH=1`,

- `NDVR_X=256`,

- `x_initial=-16.0d0` and `x_final=+16.0d0`,

- `Time_Begin=0.0d0`,

- `Time_Final=20.0d0`,

- `Integration_Stepsize=0.1d0`,

- `Write_ASCII=.T.`,

- `Coefficients_Integrator='DAV'`,

- `whichpot="HO1D"`,

- `parameter1=1.d0` (and set all the other potential parameters to zero).

Additionally, for bosons set

- `Interaction_Type=0`,

- `which_interaction='delta'` (this is the contact interaction),

- `Interaction_Parameter1=0.0d0`,

- `Interaction_Parameter2=0.0d0` (and set all the other interaction parameters to zero),

while for fermions set

- `Interaction_Type=4`,

- `which_interaction='regC'` (this is the regularized Coulomb interaction),

- `Interaction_Parameter1=0.01d0` (this parameter corresponds to $\alpha$),

- `Interaction_Parameter2=100.0d0` (this parameter corresponds to $\beta$),

- `Interaction_Parameter3=0.0d0` (set also all the other interaction parameters to zero).

Now, run your imaginary time propagation (relaxation) with `./MCTHDX_gcc`. The computation should last a couple of minutes. Fig.12 shows the results for density and trapping potential. The ground state energy is around $E_0 = 5.367407139871924$. Fig. 13 shows similar results, but for the fermionic problem (in the figure we additionally compare the fermions with regularized Coulomb repulsions to noninteracting fermions showing how the density spreads out further for the repulsive case - you do not need to do that). The ground state energy for the interacting fermions is around $E_0 = 24.86279037504089$.

## 4.2 Time evolution

Now we are ready to propagate the simulated initial state using a time-dependent Hamiltonian. The dynamics is explored in a time-dependent confining potential; a harmonic trap with a time-dependent central barrier that is slowly quenched from zero to a significant value and is given as

$$V(x;t) = \frac{1}{2}x^2 + V_0(t)e^{-2x^2}\,, \tag{22}$$

where the time-dependent barrier height is of the form

$$V_0(t) = \begin{cases} V_{\max}t/\tau\,, & t < \tau\,, \\ V_{\max}\,, & t \geq \tau\,. \end{cases} \tag{23}$$

The maximum barrier height is kept constant at $V_{\max} = 20$ for bosons and $V_{\max} = 60$ for fermions. We simulate the dynamics for various quenching rates, $\tau = 1$, 10 and 100, see Fig.14. You can find the implementation of this time-dependent double well in the file `source/ini_guess_pot/Get_1bodyPotential.F` by searching for the string `HO1D+td_gauss`.

Now, to simulate the dynamics, create a different folder for each propagation that you want to investigate. Then, copy all the binary files – namely `CIc_bin`, `PSI_bin` and `Header` – and the input file from the previous relaxation folder to the current propagation folder. After that, modify the input file `MCTDHX.inp` for a propagation as in the following inputs (keep the rest of the input file the same):

- `Job_Prefactor=(0.0d0,-1.0d0)`, which triggers forward propagation.

- `GUESS='BINR'`, which will use initial states from the binary files.

- `Binary_Start_Time=20.0d0`, which defines the starting time to be ready from the binary files. Since we let the previous relaxations run up until time step 20, this is the latest available data point (or set it according to your choice of time of relaxation).

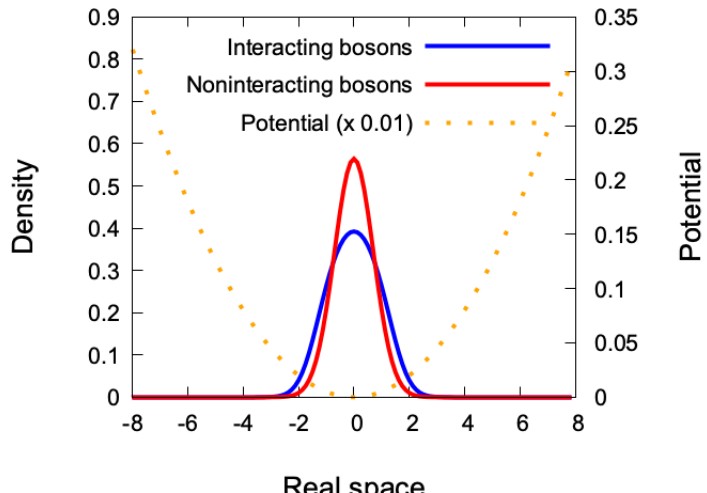

Figure 12: Many-body density for $N = 5$ bosons and $M = 10$ orbitals in a harmonic trap (orange curve, scaled by 0.01). The density for interacting bosons ($W_0 = 1$) is plotted in blue, while the density for noninteracting bosons ($W_0 = 0$) is plotted in red.

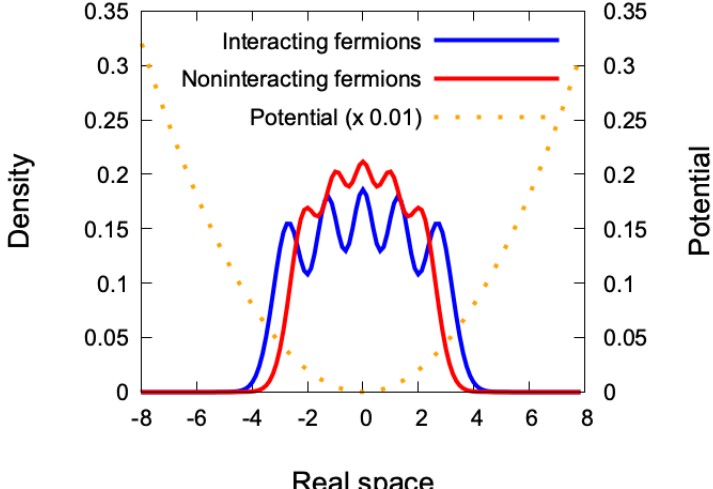

Figure 13: Many-body density for $N = 5$ fermions and $M = 10$ orbitals in a harmonic trap (orange curve, scaled by 0.01). Fermions interacting via a (repulsive) regularized Coulomb interactions (blue) are compared with noninteracting fermions (red).

- `Time_Final=100.0d0` (or you can simulate for longer times depending on the system).

- `Output_TimeStep=0.1d0` (or smaller resolution, depending how often you want to generate data; we want to be able to plot the evolution of the density as time evolves).

- `Integration_Stepsize=0.01d0` (integration of a real-time differential equation is more prone to accumulation errors, so we need to reduce the step size as compared to relaxations).

- `Coefficients_Integrator='MCS'` (we require a different integrator for propagations).

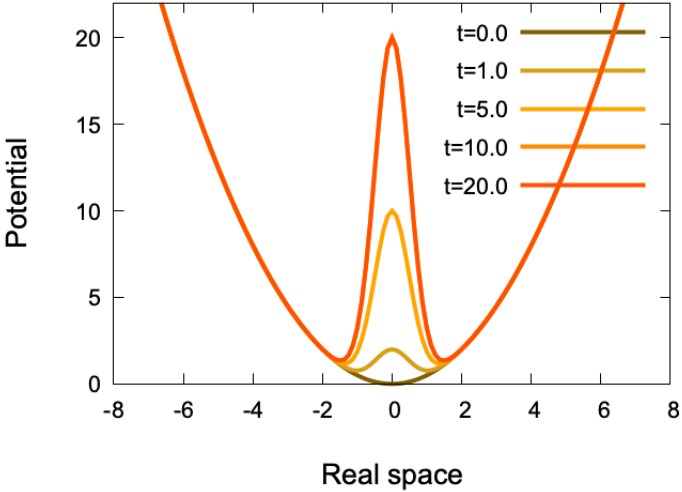

Figure 14: Quenching of the central barrier in the time-dependent potential, see Eq. 23 for quench rate $\tau = 10$. Notice that beyond time $t = 10.0$, the barrier has reached its maximal value and will stay there.

- `whichpot="HO1D+td_gauss"`.

- `parameter1=1.d0` (harmonic trap depth).

- `parameter2=0.0d0` (harmonic trap displacement).

- `parameter4=1.0d0` or `parameter4=10.0d0` or `parameter4=100.0d0` (quench rate $\tau$).

- `parameter5=0.0d0` (displacement of the Gaussian barrier).

- `parameter7=0.0d0` (and set all the other potential parameters to zero).

Additionally, for bosons set

- `parameter3=20.0d0` (maximal barrier height),

- `parameter6=0.5d0` (standard deviation of the Gaussian barrier),

while for fermions set

- `parameter3=60.0d0` (maximal barrier height),

- `parameter6=1.0d0` (standard deviation of the Gaussian barrier).

We need a higher barrier for fermions because they tend to populate higher energy states due to the Pauli exclusion principle, whereas bosons can coexist in the same energy state even with some moderate repulsion.

Finally, run `./MCTDHX_gcc` to execute your simulations. The computations should take around 30 minutes each. If you have a powerful computing station, you could try to run at least two in parallel, but they will be slower.

## 4.3 Analysis of propagation

Now, let us analyze the results of the time evolution at different propagation times and compare them across different values of the quench parameter, $\tau$. We will investigate density, natural occupations (a tool to measure the degree of fragmentation) and finally energy as a function of real time. We will also check the convergence of the propagation. We will discuss bosonic and fermionc computations separately.

### 4.3.1 Bosons

**Density**  To visualize the density, we use the `*orbs.dat` files, as already explained in the previous sections. Figs. 15, 16 and 17 depict the time evolution of the density and one-body confining potential at different time snapshots for the fast quench with $\tau = 1$ (fast quench), $\tau = 10$ (intermediate quench), and $\tau = 100$ (slow quench), respectively.

We begin by describing the dynamics of the fast quench ($\tau = 1$) in Fig. 15. At $t = 0$, the one-body potential is a single harmonic well and the density is centered around the origin in a characteristic Thomas-Fermi shape of an inverted parabola. As the propagation starts (small time $t = 0.1$), a small central barrier is raised, but no visible change in the density can be observed. At $t = 1$, however, the central barrier has reached a significant height, thereby creating an effective 1D double well. The density responds dynamically by splitting into two lobes to the left and right of the barrier. The lobes remain localized in the two wells for larger times $t = 10, 50, 100$. However, the density develops more complicated patterns as the bosons readjust to the new potential landscape. Note that the slight asymmetry displayed

at later times is a consequence of small accumulated errors in the integrator when dealing with such a strong quench, because the potential is not breaking the inversion symmetry with respect to $x = 0$. This can be corrected by employing finer integration step sizes.

The behavior in the intermediate and slow quenches differs strongly from the one in the fast quench. Already for $\tau = 10$ (Fig. 16), we observe a much smoother transition from the Thomas-Fermi profile to the two-peak structure at either sides of the barrier. In the intermediate quench, there are some slight oscillations in the height of the peaks after the barrier has reached its maximal height (cf. $t = 10$, $t = 50$, $t = 100$). This indicates that there is still some internal density reconfiguration happening as a remnant of the quench procedure.

In the slow quench (Fig. 17), instead, the double peak configuration is reached in a very smooth way with minimal disturbance. Here, we are approaching an adiabatic limit and thus the initial ground state should be mapped to the ground state of the Hamiltonian at the final time, in accordance with the adiabatic theorem [69].

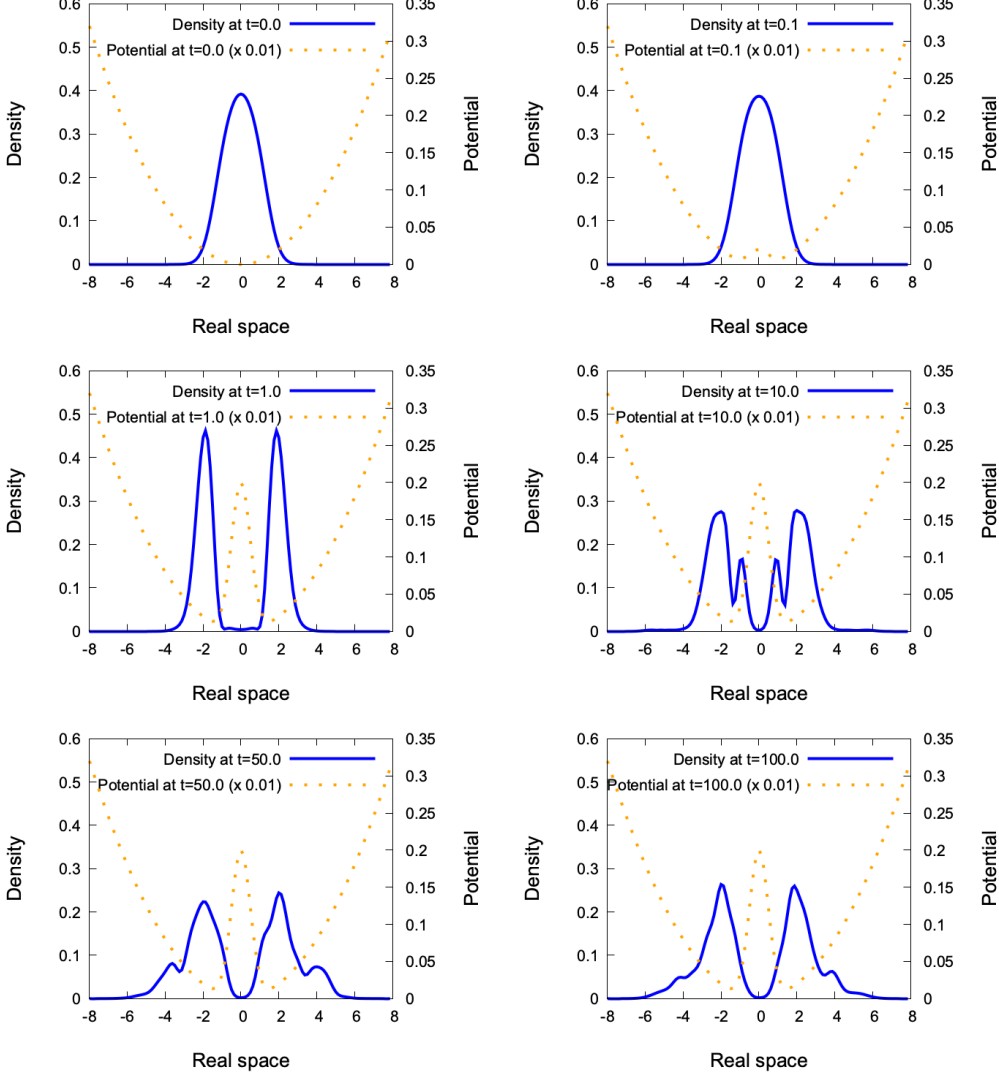

Figure 15: Behavior of the density (solid blue) and one-body potential (dotted orange) as a function of time for 5 bosons in a double well whose central barrier is quenched at a rate $\tau = 1$ (very fast quench).

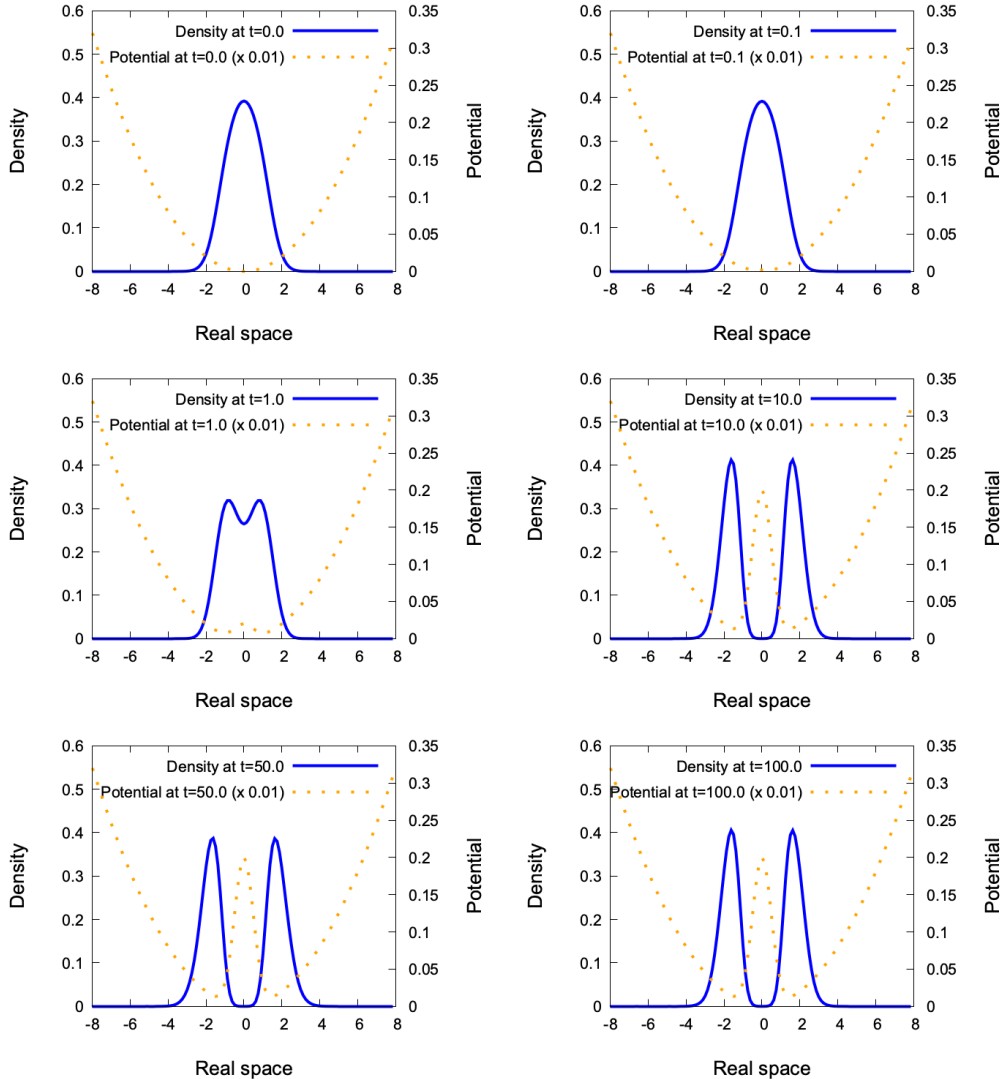

Figure 16: Behavior of the density (solid blue) and one-body potential (dotted orange) as a function of time for 5 bosons in a double well whose central barrier is quenched at a rate $\tau = 10$ (intermediate quench).

**Natural occupations**  The dynamics of the natural occupations $n_i(t)$ are plotted using the file `NO_PR.out`. Fig. 18 shows the time evolution of these quantities for the three different choices of $\tau$. Recall that, to simulate the dynamics, we employed $M = 10$ self consistent orbitals. In the figure, though, only three natural occupations, $n_1/N$, $n_2/N$ and $n_3/N$ are shown, since all the other orbitals have much smaller occupations than the third (the occupations are saved from the smallest to the largest in `NO_PR.out`). In all cases, we can appreciate that the system starts from a superfluid state with a single occupied orbital. The subsequent occupation dynamics depends strongly on the quench rate $\tau$.

For the fast quench with $\tau = 1$, depletion of the initial orbital occurs almost immediately and the condensate evolves into a nearly three-fold fragmented state at large times. The dynamics is not very regular as suggested already by the behavior of the density.

For the intermediate quench with $\tau = 10$, the condensate depletion is visibly reduced. Only the second orbital acquires a sizeable population over time, while the third orbital occupation remains close to zero. Moreover, we observed regular oscillations in the orbital occupations which mirror the oscillations in the density peaks. Nevertheless, the system approaches a

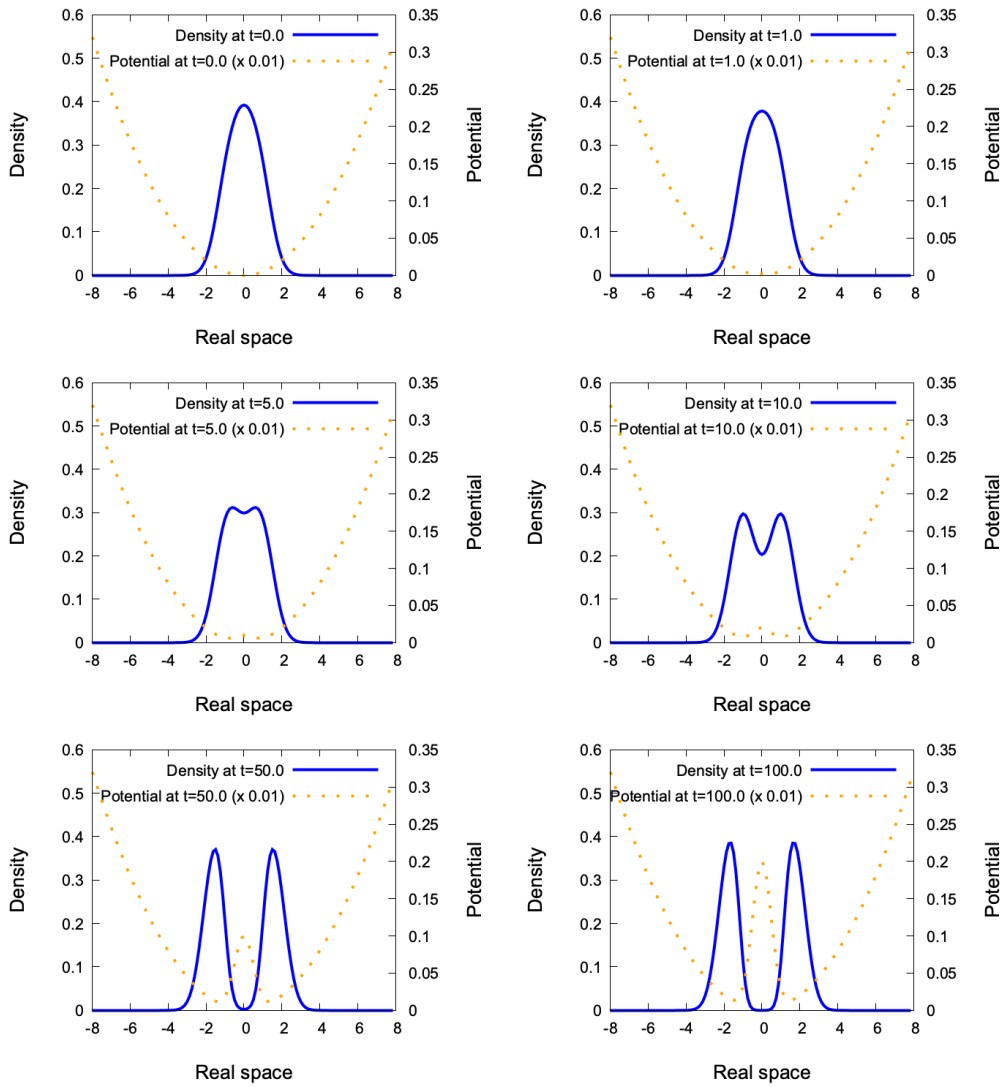

Figure 17: Behavior of the density (solid blue) and one-body potential (dotted orange) as a function of time for 5 bosons in a double well whose central barrier is quenched at a rate $\tau = 100$ (very slow quench).

steady-state with around 80% occupation in the largest orbital and 20% in the second-largest. This oscillation can be understood in a effective two-state model. In the transient time explored here, the many-body state oscillates between two different configurations: one with a larger superfluid component (peaks of first orbital, valleys of the second), and another one with a larger fragmentation (valleys of the first orbital, peaks of the second). If you compare the orbital occupation with the density, you can see that the first configuration (more superfluid) corresponds to the case where the two density peaks are closer to one another, while the second configuration (more fragmented) corresponds to the peaks being further apart. As the many-body state settles into its steady-state configuration, the oscillations progressively dampen and the density becomes distributed in an intermediate "80/20" configuration between the two.

Finally, in the slow, almost adiabatic quench with $\tau = 100$, we observe a much smoother transition to the same 80/20 steady state exhibited in the intermediate quench. The oscillations, though, have a much smaller amplitude in this case.

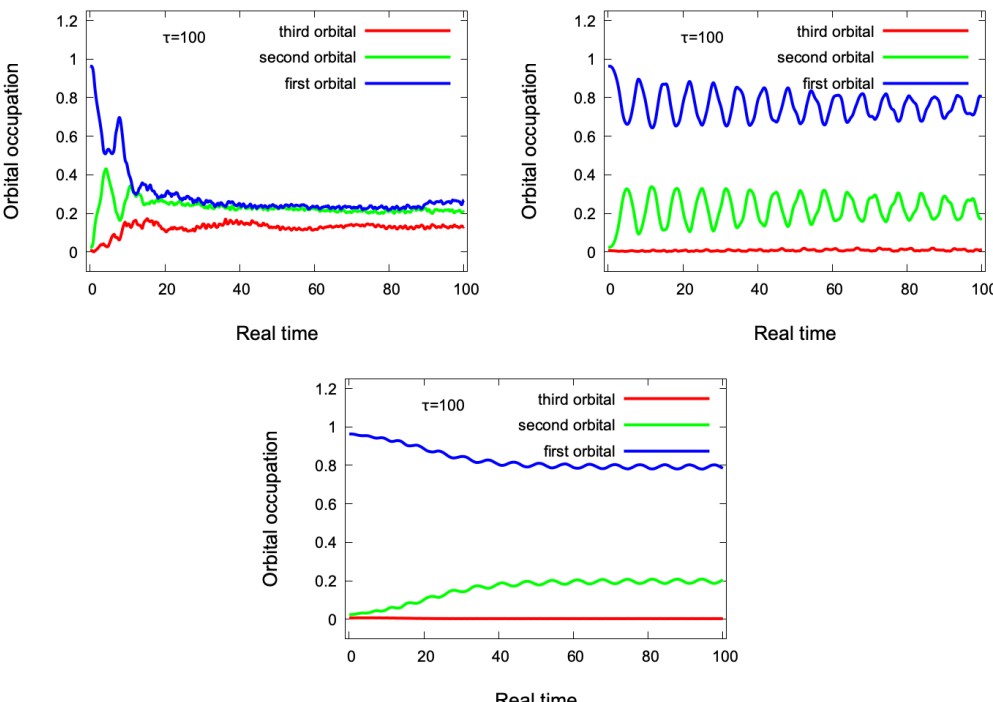

Figure 18: Time evolution of the three leading natural occupations for $N = 5$ bosons in a double well quenched with different rates $\tau$.

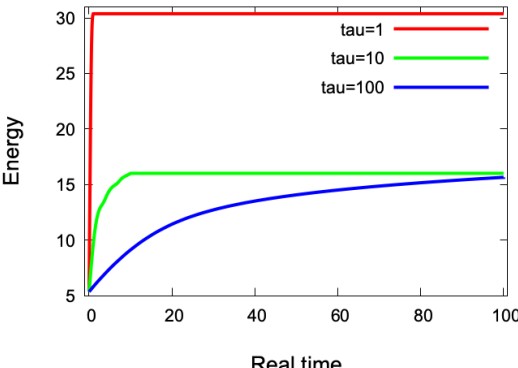

Figure 19: Energy convergence after the quench procedure with $N = 5$ bosons and $M = 10$ orbitals for different values of the quench rate $\tau$.

**Energy** Since we are changing the parameters of the Hamiltonian by progressively raising the central barrier, we expect the energy to change as a function of time. Intuitively, the energy of the propagated state in the long time limit should be higher than the ground state energy of the initial configuration with no barrier, because the barrier forces the particles to sit at a higher potential energy. We can check our intuition by plotting the energy dynamics from the `NO_PR.out` file, shown in Fig. 19 for $\tau = 1$, 10 and 100. For all values of $\tau$, the energy saturates in the steady state. Interestingly, the intermediate and slow quench bring the system to stead states with very similar energy (and as we saw with very similar density). The fast quench ($\tau = 1$), instead, introduces twice as much energy into the system by forcing the particles further away from the center of the potential.

### 4.3.2 Fermions

We now perform an analysis for similar time-dependent computations involving fermions.

**Density**    In Figs. 20,21, and 22 we plot some snapshots of the density for the different quenching rates. For this, we again employ the *orbs.dat as described in the relaxation section.

As already seen for the bosonic case, when $\tau = 1$, we are performing a fast quench which is known to introduce a lot of energy into the system and drastically reorganize the density in time. In fact, we see a very turbulent dynamics with a lot of oscillations within the two wells once the barrier is raised. Note that, while turbulent, the density remains confined in the double well trap. You can see if you plot it in the full spatial grid, $x = -16$ to $x = +16$. In Fig. 20, we restricted the visualization to the region $x = -8$ to $x = +8$ to better capture local density details.

As we slow down the quench rate ($\tau = 10$), the dynamics becomes smoother and the steady state is more stable. Due to the different quantum statistics, the overall density configuration is quite different than the one exhibited by bosons. Within each peak in both wells, we can observe sub-peaks which are the product of the Pauli exclusion principle. Since the quench is moderate, the density continues to oscillate between different configurations before settling to a steady state. Interestingly, in this case there is no simple two-state picture appearing. Instead, as we will see also for the occupation, there are at least two types of oscillations at play. A first, faster oscillation is similar to the one observed for bosons, where the center of mass of each peak oscillates closer to and further away from the trap center. A second, slower beating pattern appears on top of the faster one, and shows the two density clusters in each well oscillating between a two-peak and a three-peak density profile configuration. This is a manifestation of a complex interplay between long-range Coulomb interactions and the frustration introduced by having an odd number of particles in a two-fold symmetric structure.

Finally, for the slowest quench procedure ($\tau = 100$), the dynamics is quasiadiabatic as already seen for bosons, and the final fragmented state is reached very smoothly.

**Orbital occupation**    We also plot the natural occupations $n_i(t)$ using the file NO_PR.out for the quenched fermions. As already seen for bosons, another indication of the very different behavior between fast and slow (adiabatic) quenches is given by the orbital occupation. Due to the different quantum statistics, when dealing with fermions we require at least $M = N + 1$ orbitals as the first $N$ will always be populated. Since in this case $N = 5$, it is thus most instructive to plot and compare the natural orbital occupations of the fifth and sixth highest orbital for the different ramp up times. These are found in the sixth and fifth column of NO_PR.out, respectively, and are shown in the left and right panel of Fig. 23.

For $\tau = 1$, i.e. when the barrier is raised very rapidly, the many-body state does not have time to adapt to the quenched configuration and therefore many more orbitals become populated. In particular, the population of the fifth orbital drops rapidly from 20% (indicating one orbital per particle) and stays low throughout the time evolution. Simultaneously, the next orbital (sixth) becomes macroscopically populated.

For the intermediate quench at $\tau = 10$, the fifth orbital maintains at first the initial 20% occupation, but eventually it drops slightly with the same double-oscillation pattern observed in the density. Concomitantly, the sixth orbital acquires an occupation of about 2%, mirroring the same kind of oscillations. This is an indication that the quench is fast enough to perturb the state into a transient (slight) nonequilibrium configuration before the steady state sets in. Note that the simple two-state picture does not apply here. You can check this yourself by plotting the remaining orbitals and verifying that they acquire an occupation in the same order of magnitude (seventh orbital: 1%, eight orbital: 1%, ninth orbital: 0.5%, tenth orbital: 0.2%).



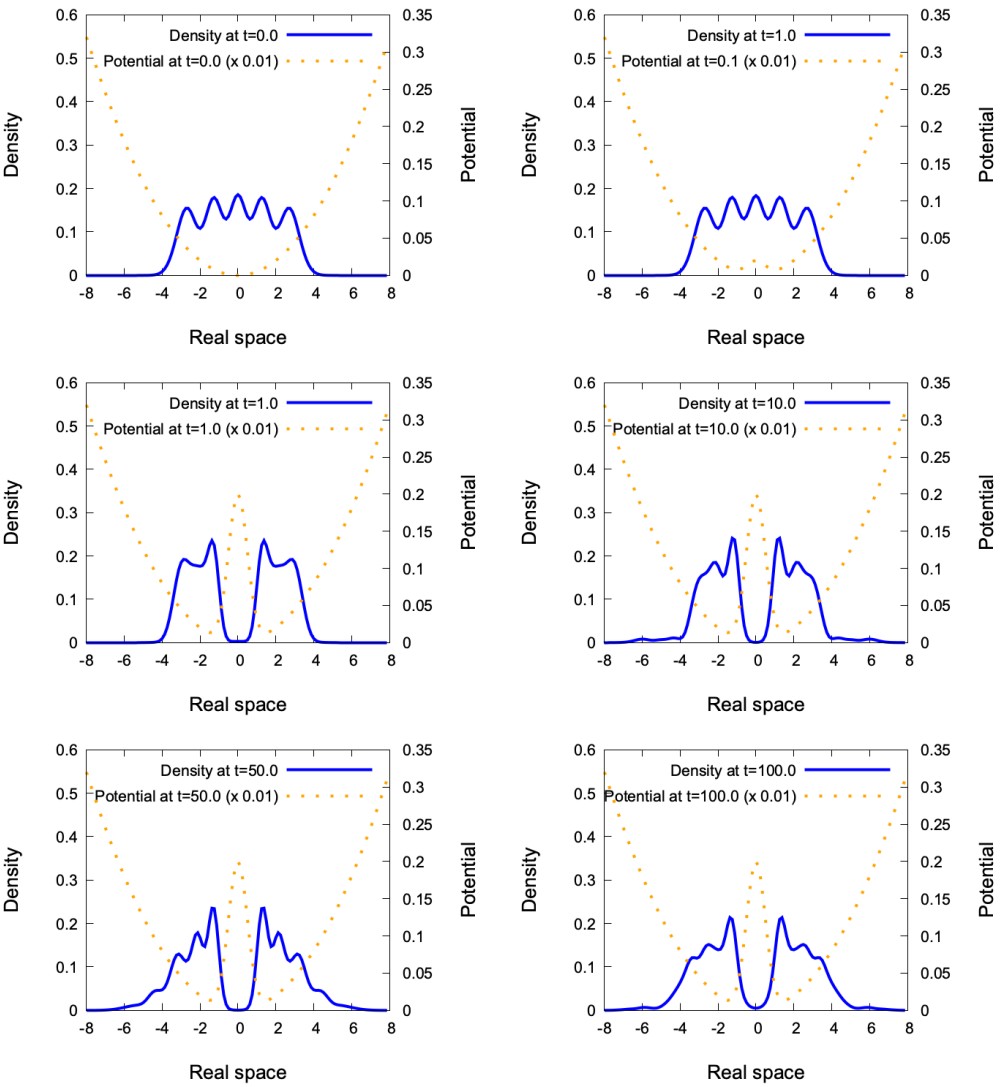

Figure 20: Behavior of the density (solid blue) and one-body potential (dotted orange) as a function of time for 5 fermions in a double well whose central barrier is quenched at a rate $\tau = 1$ (fast quench).

Finally, for $\tau = 100$, we observe a nice adiabatic transition from the initial ground state to the ground state of the final potential, indicated by the practically constant orbital occupation of the fifth orbital at 20% and of the sixth orbital near zero.

**Energy** The distinction between adiabatic and diabatic quenches can also be seen in the energy in Fig. 24. The energy distinguishes two regimes: the fast quench where the final state is a highly excited state of the final double-well Hamiltonian, and the slow (quasi-adiabatic) quench where we converge to a low-energy state of the final double well (strictly speaking, the ground state is obtained only for an infinitely slow quench $\tau \to \infty$).

## 4.4 Additional exercises

**What is the true ground state?** It is interesting to note that the symmetric low-energy state in the fermionic problem does not actually match the ground state that we would obtain for the double well potential from a relaxation. This is because the system is highly frustrated with

the given odd number of fermions ($N = 5$) for an even number of wells (2). Let us elaborate on what we mean with frustration. The first 4 fermions can sit symmetrically in localized pairs on either side of the wall. The fifth fermion, however, faces a dilemma. To respect the axial symmetry of the potential, it should delocalize across the two wells. However, to minimize the energy contribution from the one-body potential, it would be best to choose one of the two wells to sit in. This, in turn, leads again to an energy penalty due to the Pauli exclusion principle, as we would be cramming three fermions in the same well.

In our initial simulations, it appears that the symmetry-broken configuration is preferred in the relaxation. We can verify this by performing a relaxation in the double-well starting from the ground state of harmonic potential without barrier, see Fig. 25. At intermediate imaginary times, we indeed converge to the symmetric state seen in the adiabatic propagations. At longer imaginary times, however, the algorithm suddenly unlocks a pocket at lower energy with a different and asymmetric density distribution with roughly two particles in one well, and three in the other. In order to plot those energies, follow the instructions from the ground state part of the tutorial.

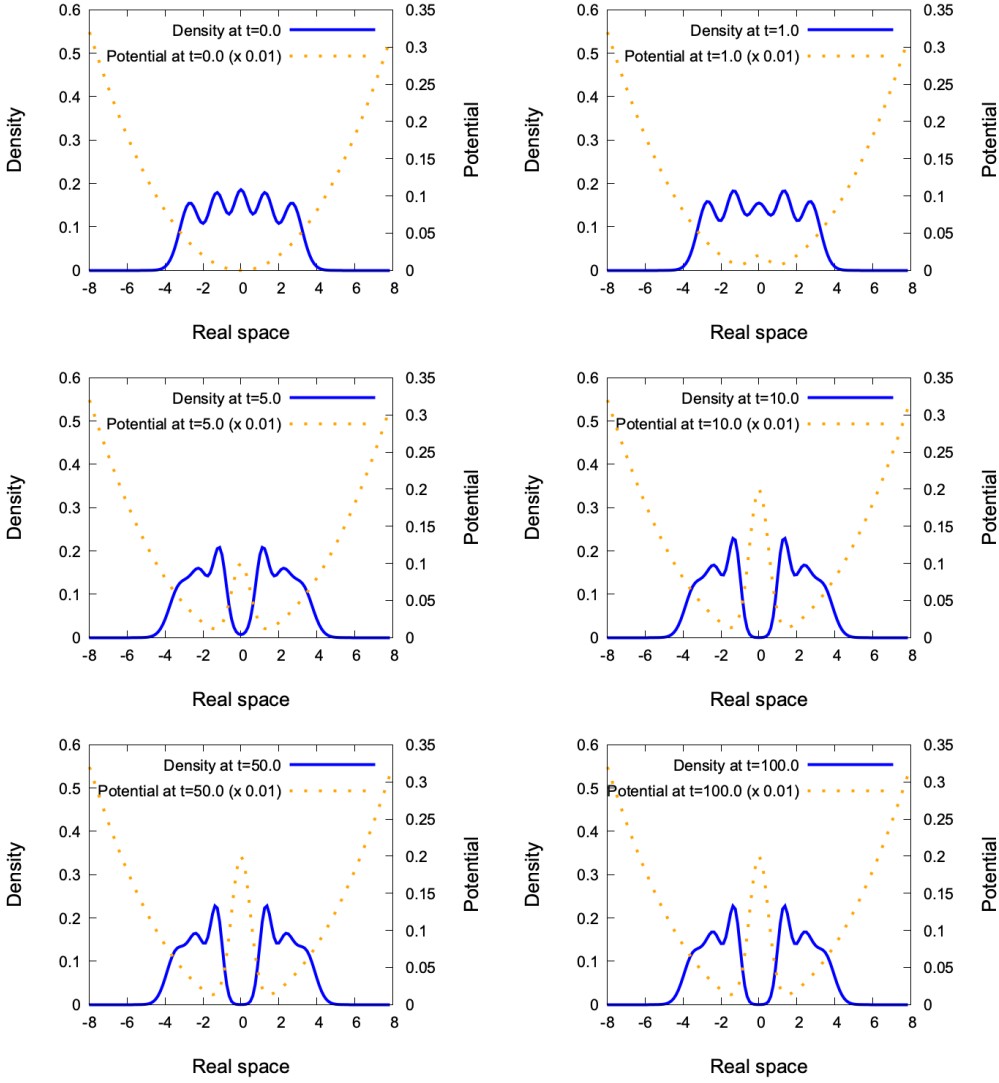

Figure 21: Behavior of the density (solid blue) and one-body potential (dotted orange) as a function of time for 5 fermions in a double well whose central barrier is quenched at a rate $\tau = 10$ (intermediate quench).

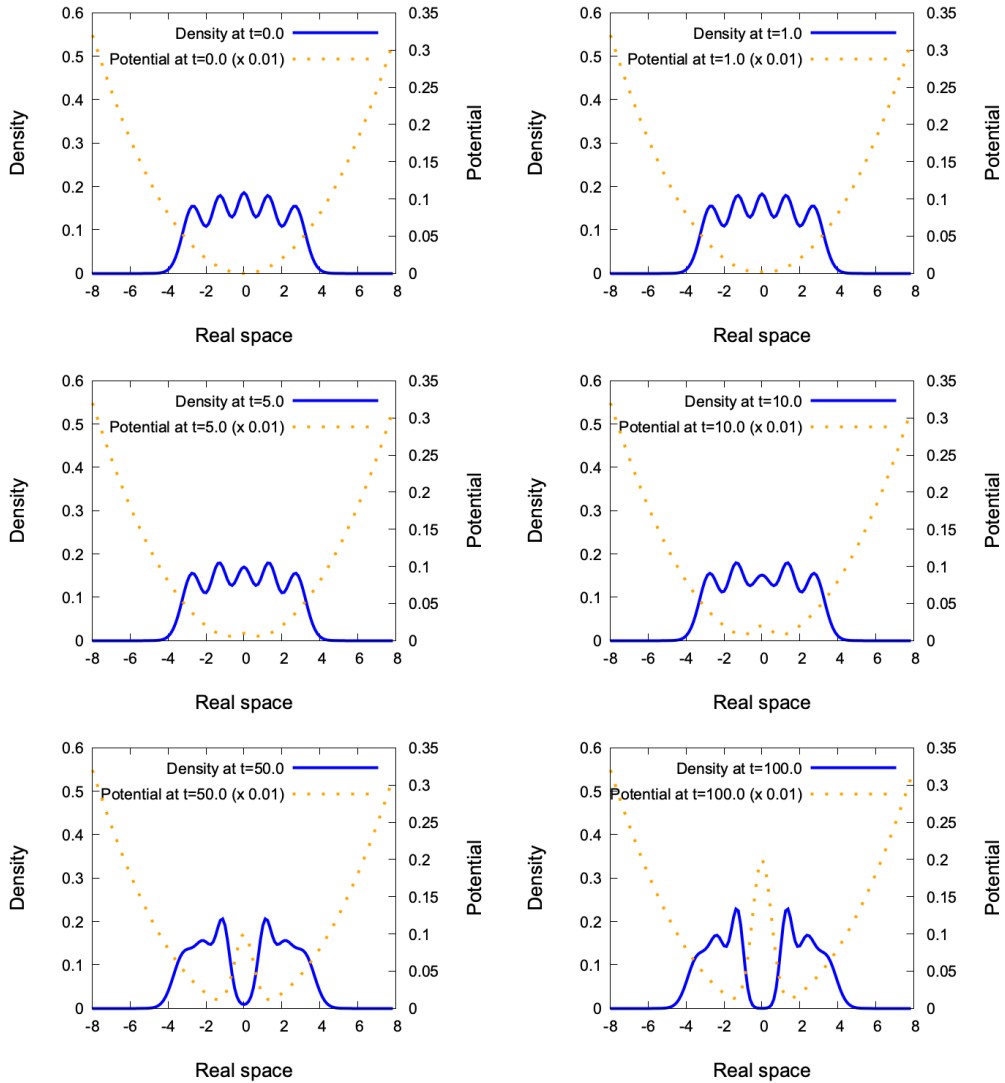

Figure 22: Behavior of the density (solid blue) and one-body potential (dotted orange) as a function of time for 5 fermions in a double well whose central barrier is quenched at a rate $\tau = 100$ (very slow quench).

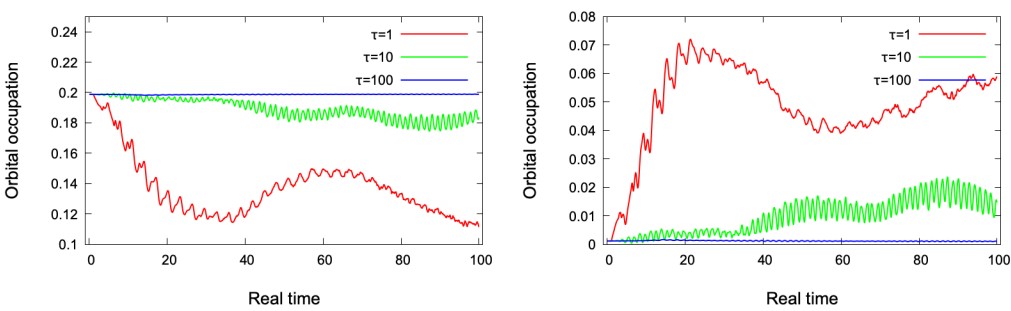

Figure 23: Time evolution of the fifth (left) and sixth (right) leading natural occupations for $N = 5$ fermions with $M = 10$ orbitals in a double well quenched with different rates $\tau$.

Why is the relaxation suddenly breaking the axial symmetry $x_j \rightarrow -x_j$, which is ingrained in the Hamiltonian through the one-body potential and the interparticle interaction? Symmetry-broken solutions emerge when we have non-linear equations of motions and indicate that the many-body solution is particularly interesting, in this case due to frustration. In fact, the linear combination of the symmetry-broken solution and its mirror image will be (very slightly) lower in energy when sandwiched with the (true, linear) many-body Hamiltonian. This problem happens also in attractive bosons in a 1D double well, and already for $M = 1$. The symmetry broken $M = 1$ Gross-Pitaevskii solution (localized in one of the two wells) becomes lower in energy than the $M = 1$ symmetry preserving solution (localized in both wells). This indicates that one needs more orbitals to achieve the true ground state. In the bosonic case, $M = 2$ or $M = 4$ will get the right ground state. This state is a "cat state", a quantum superposition of two symmetry-broken states. It is called a cat state because it is conceptually similar to Schrödinger's cat state: a superposition of two macroscopically distinct configurations, such as all particles localized in one well versus all in the other.

The cat state is inherently fragile due to its delicate energy balance and sensitivity to perturbations. In nonlinear systems, numerical algorithms often converge to symmetry-broken states, which are local energy minima, rather than the true ground state represented by the cat state. Limited orbital space in simulations further compounds this issue, as it cannot capture the necessary many-body correlations. Additionally, even in real experiments small perturbations or interactions with the environment can cause decoherence, collapsing the superposition into one of the symmetry-broken states. Achieving the true cat state requires careful control, enhanced orbital resolution, and sophisticated numerical approaches.

How can we fix this? The equations of motion need a large enough number of degrees of freedom (i.e. orbitals) to boil down to the true ground state – see the above example for attractive bosons and $M = 1$. It is simpler to see this process at play for a double well with only $N = 3$ fermions, and for a smaller barrier of height 20 and width 0.5. With lower values of $M$ (e.g. $M = 6$ or $M = 12$), you will get the symmetry broken solution. For $M = 18$, you can manage to restore the symmetry, as shown in Fig. 26. The critical value of $M$ for which the symmetric ground state is restored depends on the interaction strength, the initial shapes of the orbitals, the splitting between the two quasi-degenerate ground and first excited states, and how the numerics is able to handle all of these factors.

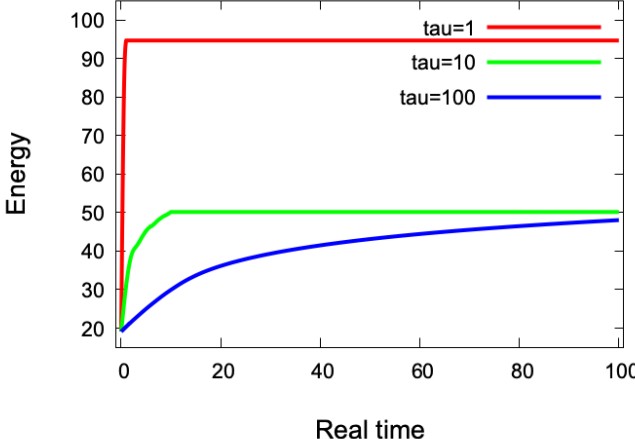

Figure 24: Energy convergence after the quench procedure with $N = 5$ fermions and $M = 10$ orbitals for different values of the quench rate $\tau$.

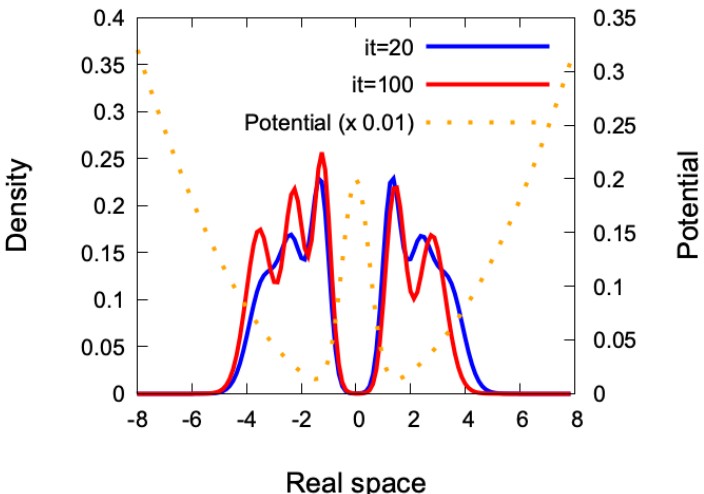

Figure 25: Quasi-degenerate states of $N = 5$ fermions with $M = 10$ orbitals in a double well: ground state (red curve) and excited state (blue curve, similar to the long-time density of an adiabatic quench as in Fig. 22).

Note that the convergence to the ground state is particularly slow for $M = 18$. We reach the symmetric state only around time $t = 160$! One approach to deal with such difficult problems is to start with a smaller grid (e.g. 64 grid points instead of 128) and then interpolate and re-run. MCTDH-X can handle automated grid interpolation on its own: it detects whether the grid has been increased from a previous computation, and performs an interpolation for the higher-resolution grid. Indeed, we can see in Fig. 27 that for 64 grid points the convergence is much faster, and at $t = 40$ we already reach the symmetric state. It is then straightforward to redo a calculation with the larger grid (128 points) using the initial state of the 64-point grid calculation and verify that the symmetry is preserved, as MCTDH-X automatically interpolates initial states defined on a coarser grid.

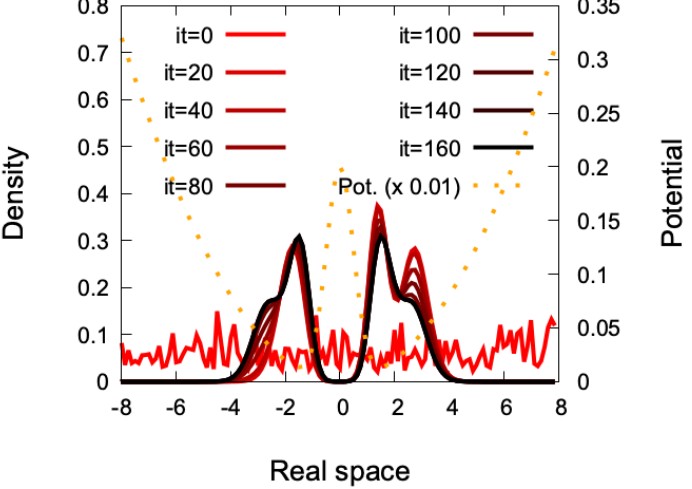

Figure 26: Imaginary time relaxation to the symmetric ground state of $N = 3$ fermions in a double well with $M = 18$ orbitals and 128 grid points.

# 5 Advanced features

## 5.1 Analysis

So far we have seen how MCTDH-X can calculate bosonic and fermionic ground state properties such as energy and orbital occupation, and perform real time evolution of the many-body state to generate a time-dependent real-space density. Another important feature is the ability to calculate correlation functions and other observables. This is performed with a separate program execution *after* we have relaxed or propagated the many-body state. The executable in question is `MCTDHX_analysis_gcc`, which is always imported when you run the command `bincp`. This executable has its own corresponding input file, `analysis.inp`, which is imported with the command `inpcp`. The analysis program takes the binaries and ASCII files calculated during the relaxation or propagation, and extracts observables from the many-body state such as momentum-space densities, Glauber correlation functions, variances, single-shot measurements, and more.

Let us briefly have a look at the variables that can be configured in the analysis input file. We will only focus on the basic ones to generate energies, density distributions, and correlations.

- `Total_Energy`: if set to true (`.T.`), calculates a breakdown of the total energy in kinetic, potential, and interaction energies.

- `Time_From`: this variable tells the program at which time to start performing the analysis. All the time steps from this time until the final time (`Time_To`) will be evaluated.

- `Time_To`: this variable tells the program at which time to end the analysis.

- `Time_Points`: this variable counts the number of time points to perform the analysis at. E.g. if you want all integer times from 0 to 50 included, you need to set `Time_From=0.0d0`, `Time_To=50.0`, and `Time_Points=51`. Note that these times can refer to both a relaxation or a propagation.

- `Density_x`: if set to `.T.`, the program will (re)-calculate the real-space density for all the requested times.

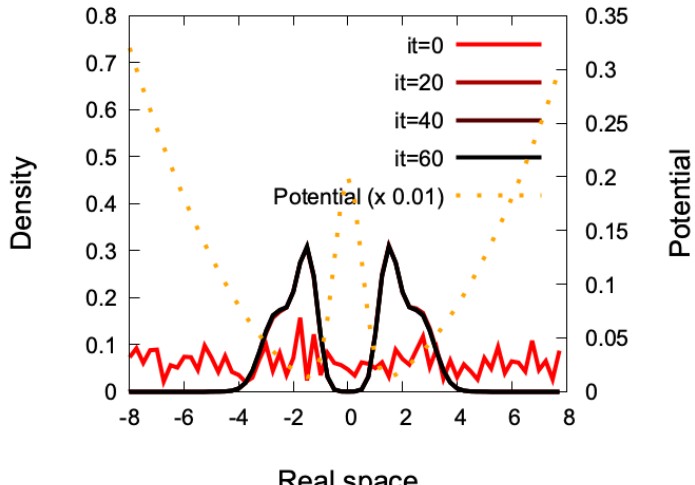

Figure 27: Imaginary time relaxation to the symmetric ground state of $N = 3$ fermions in a double well with $M = 18$ orbitals and 64 grid points.

- `Density_k`: if set to `.T.`, the program will calculate the (diagonal of the one-body) density in k-space for all the requested times.

- `Correlations_X`: if set to `.T.`, the program will calculate the reduced one-body and (the diagonal of) the two-body density matrix in real space:

$$\rho^{(1)}(x, x') = \langle \Psi | \hat{\Psi}^{\dagger}(x) \hat{\Psi}(x') | \Psi \rangle \,, \tag{24}$$

$$\rho^{(2)}(x, x') = \langle \Psi | \hat{\Psi}^{\dagger}(x) \hat{\Psi}^{\dagger}(x') \hat{\Psi}(x') \hat{\Psi}(x) | \Psi \rangle \,. \tag{25}$$

These can be used to calculate Glauber one-body and two-body correlation functions as

$$g^{(1)}(x, x') = \frac{\rho^{(1)}(x, x')}{N \sqrt{\rho(x)\rho(x')}} \,, \tag{26}$$

$$g^{(2)}(x, x') = \frac{\rho^{(2)}(x, x')}{N^2 \rho(x)\rho(x')} \,. \tag{27}$$

- `Correlations_K`: if set to `.T.`, the program will calculate the reduced one-body and (diagonal of the) two-body densities in k-space.

## 5.2 Correlations for bosons and fermions

Armed with the knowledge above, we will run our own analysis and calculate one-body and two-body correlation functions for some of the above bosonic and/or fermionic systems.

We will first use the analysis program `MCTDHX_analysis_gcc` to create the files required for plotting the one-body and two-body correlation functions in space.

In the directory of your dynamics calculation, first change the following parameters in the analysis input file `analysis.inp`. If this is not already in your current folder, you can copy it from your relaxation directory.

- `Time_From=0.0d0`: We want to analyze the system from the beginning of the propagation at time $t = 0$.

- `Time_To=100.0d0`: We want to analyze the system until the end of our propagation.

- `Time_Points=101`: We want to evaluate the correlation functions for 101 time steps starting with 0 and ending with 100.

- `xstart = -16.0d0`: This is the beginning of the grid.

- `xend = 16.d0`: This is the end of the grid.

- `Correlations_X=.T.`: Setting this flag to true will calculate the one-body and two-body correlation functions in real space.[5]

Then we run the analysis program from the terminal using

```
./MCTDHX_analysis_gcc.
```

For the case of the above fermionic dynamics calculations, this produces the files `*N4M8x-correlations.dat`. These files contain the values of the one-body and two-body reduced density matrices at different positions $x$ and $x'$. For example, the real and imaginary

---

[5] Setting `Correlations_K=.T.` will calculate the one-body and two-body correlation functions in *momentum* space, which is not discussed here.

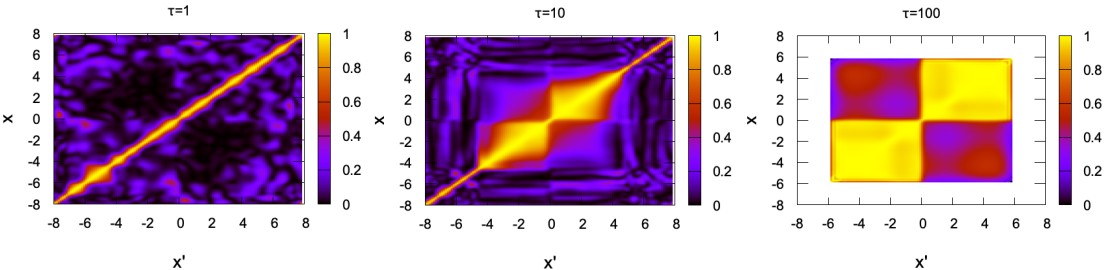

Figure 28: Real-space one-body correlation function $g^{(1)}(x, x')$ at time $t = 99$ for different quench rates $\tau$ in the bosonic double-well quench.

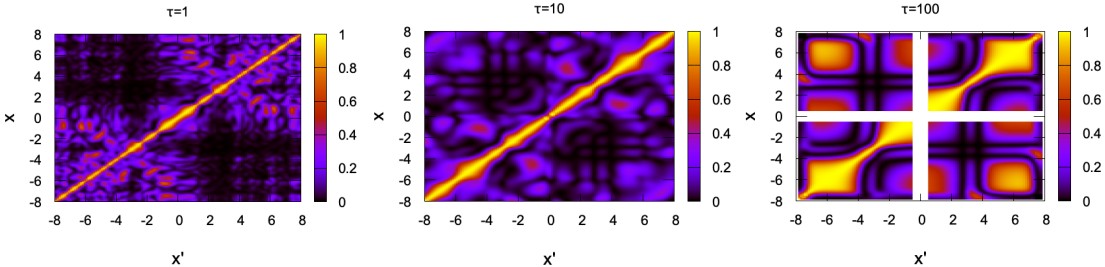

Figure 29: Real-space one-body correlation function $g^{(1)}(x, x')$ at time $t = 99$ for different quench rates $\tau$ in the fermionic double-well quench.

part of $\rho^{(1)}(x, x')$ are saved in the 8th and 9th column, while the 7th and 10th column contain $\rho(x)$ and $\rho(x')$, respectively. Please refer to the manual for a full description of the correlation files.

Now that all our necessary files are generated, we will plot the correlation functions in real space. For this, open gnuplot and execute the following lines one by one:

```
set pm3d map,
set cbrange [0:1],
splot "99.0000000N5M10x-correlations.dat" u 1:4:(sqrt($8**2 + $9**2)/sqrt($7*$10)).
```

The lines above will display a plot of the real-space one-body correlation function. For the two-body correlation function, type the following:

```
set pm3d map,
set cbrange [0:1],
splot "99.0000000N5M10x-correlations.dat" u 1:4:($11/($7*$10)).
```

The one-body correlation function $g^{(1)}(x, x')$ at time $t = 99$ and for different ramp up times is illustrated in Fig. 28 for the bosonic setup and in Fig. 29 for the fermionic one. Note that white areas represent artificial numerical singularities due to $\rho(x)$ or $\rho(x')$ being zero.

Different points of $g^{(1)}(x, x')$ can be interpreted as a measure of the coherence. If $g^{(1)}(x, x')$ is large, the quantum states at $x$ and $x'$ are highly coherent and can exhibit strong interference effects. In contrast, if $g^{(1)}(x, x')$ is small or zero for large separations, it indicates that the system lacks long-range coherence, typical of normal fluids or gases. As a special case, wherever the probability is non-zero along the main diagonal, the probability must be one, because every state is perfectly coherent with itself.

Examining the one-body correlation for the bosonic and fermionic problems, we can see that in the fast quench ($\tau = 1$) coherence between the two wells is low, because the particles did not have enough time to adapt to the fast separation of the system into two distinct wells.

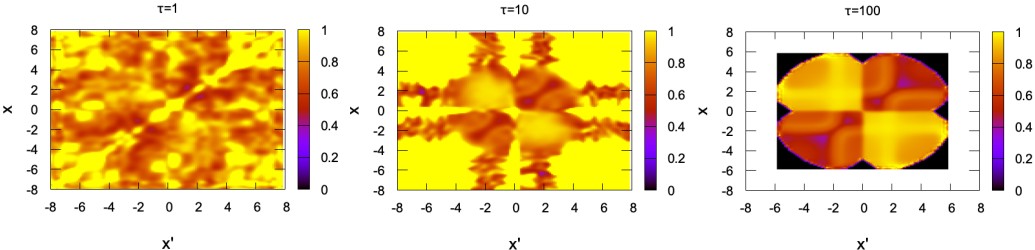

Figure 30: Real-space two-body correlation function $g^{(1)}(x, x')$ at time $t = 99$ for different quench rates $\tau$ in the bosonic double-well quench.

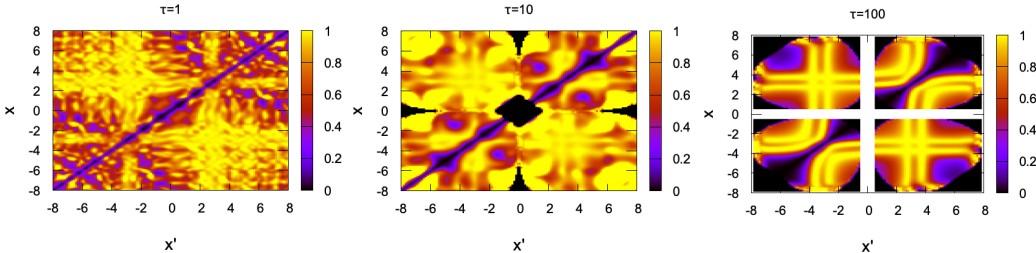

Figure 31: Real-space two-body correlation function $g^{(1)}(x, x')$ at time $t = 99$ for different quench rates $\tau$ in the fermionic double-well quench.

As the barrier is raised more slowly, there is more correlation between the two wells as evinced by a larger off-diagonal $g^{(1)}(x, x')$, e.g. between $x = 2$ and $x' = -2$ in the bosonic case.

In the adiabatic limit, we encounter the strongest off-diagonal correlation. The particles are well-localized into two separate peaks, but they retain some correlation across the wells. This is particularly striking for fermions, where we see strong off-diagonal one-body correlation between say $x = 6$ and $x' = -6$. The persistence of correlations across the two wells is evidence of quantum tunneling.

The two-body correlation function $g^{(2)}(x, x')$ at time $t = 99$ and for different ramp up times is illustrated in Fig. 30 for the bosonic setup and in Fig. 31 for the fermionic one. The two-body correlation function measures the joint probability of finding a pair of particles at positions $x$ and $x'$ simultaneously. In bosonic systems, particles tend to bunch together due to Bose-Einstein statistics, leading to $g^{(2)}(x, x')$ being larger than the uncorrelated product at short distances. In fermionic systems, instead, the Pauli exclusion principle leads to anti-bunching, where $g^{(2)}(x, x')$ is smaller than the uncorrelated product at short distances. Therefore, for the (spin-polarized) fermionic two-body correlation function, the Pauli exclusion principle can clearly be seen on the diagonal. Here, the probability is zero, since two fermions cannot coexist in the same state. For electrons, this effect is known as the Fermi hole.

## 5.3 Videos

To visualize the density as a function of time, MCTDH-X also provides scripts that generate videos automatically, based on the `*orbs.dat` files. For them to work correctly, you should have `mencoder` installed in your system. When you run the MCTDH-X installer, it should have automatically detected `mencoder` and installed the program with the video generation options. If you do not have `mencoder` installed in your system, you can still create your own movies with a different script, e.g. in python.

To create a movie of the density in time, you should run

```
1D-DENSITY_X $1 $2 $3 $4 $5,
```

where \$1 is the starting time, \$2 is the final time, \$3 is the time step, \$4 is the number of orbitals, and \$5 is the number of particles. In the present case of $N = 5$ bosons or fermions and $M = 10$ orbitals with a computation running from 0 to 100 in steps of 0.1, the syntax is

```
1D-DENSITY_X 0 100 0.1 10 5.
```

If you run this script, it should automatically generate a movie called `MCTDHX_Density_X_0_100.mpg` which you can then watch to understand the behavior of the density as the potential is modified.[6]

You can also create time-resolved videos for the one-body and two-body correlation functions using

```
1D-CORR1-X 0 100 1 8 4,
```

and

```
1D-CORR2-X 0 100 1 8 4,
```

respectively. The syntax is the same as for the videos of the time-resolved density. You can also follow the video tutorial uploaded on YouTube for more information on how to plot quantities obtained from the analysis.

## 6  Linux/UNIX command cheat sheet

Table 5: Summary of useful Linux/UNIX commands when operating MCTDH-X.

| Command | Effect |
|---|---|
| `cp path/to/myfile another/path/` | Copies a file to a different location. |
| `ls` | Lists all files in the current directory. |
| `ls -a` | Lists all files in the current directory, including the hidden ones. |
| `mkdir mydir` | Creates a new directory named `mydir`. |
| `mv path/to/file another/path/` | Moves a file to a different location. If the location is the same, but the target filename is different, it renames the file. |
| `pwd` | Print the current directory. |
| `rm myfile` | Removes (deletes) the file `myfile`. |
| `rm -r mydir` | Removes (deletes) the directory `mydir`. |
| `source myscript` | Executes the content of `myscript` in the current shell. |

---

[6]If your media player does not have a .mpg decoder, you can convert it to a more common .mp4 format with a free online converter, see e.g. cloudconvert.com.

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
