# Peer review of "Lecture Notes: Many-body quantum dynamics with MCTDH-X"

_SciPost Physics Lecture Notes, doi:SciPost Phys. Lect. Notes 94 (2025)_

## Round 1 · Referee Report · Anonymous (Referee 1) · 2024-10-20

Strengths

  1. Clarity in presenting the basics of MCTDH-X theory: these lecture notes are structured to guide the reader through the Multiconfigurational Time-Dependent Hartree approach for both bosonic and fermionic systems. They clearly introduce the theoretical framework, emphasizing the contrast between MCTDH and mean-field theories like Gross-Pitaevskii or Hartree-Fock

  2. Effectiveness in guiding the reader through complicated many-body problems: the article provides comprehensive benchmarks and step-by-step workflows, including exact solutions for specific model problems (the Harmonic Interaction Model) . It not only introduces the time-independent and time-dependent cases but also illustrates how to handle complex systems using numerical methods, with a balance between practical and theoretical aspects

  3. Completeness: the coverage of both imaginary- (for ground state) and real-time dynamics, as well as advanced analysis features like correlation functions, is thorough. The authors further includes discussion of delicate topics like symmetry-breaking and restoration.

Weaknesses

  1. Presentation of results: while the paper discusses various types of results (e.g., energy, orbital occupation, density), some sections could benefit from better presentation of the results in the graphs

Report

These lecture notes provide a comprehensive guide to MultiConfiguration Time-Dependent Hartree methods for bosons and fermions, with a good selection to problems and systems that clearly demonstrate the power of the method for many-body problems involving collections of indistiguishable particles (either bosons or fermions).

The reader is guided through these problems in an incremental way, starting from time-independent problems where imaginary-time dynamics is used to relax the systems to their ground-state, and then moving to true, time-dependent problems where a time-dependent (one-body) potential is introduced to drive the system out of a trapped state. For the first problem an analytic solvable problem (the Harmonic interaction model) is used to benchmark the numerical results of MCTDH-X, helping the reader to gain confidence with numerical parameters like the primitive grids used to represent the orbitals and, above all, the number of orbitals included in the wavefunction expansion. The second set of problems, on the other hand, help introducing the reader to advanced tools like the one-body density matrix and the correlation functions, and lead him/her to advanced issues such as correlation effects and wavefunction fragmentation.

Overall, this tutorial provides a strong foundation for researchers and students interested in the MCTDH-X method and many-body physics. It is comprehensive yet approachable, making it an ideal resource for both beginners and more advanced users. The clear distinction between time-independent and time-dependent problems, coupled with practical examples and benchmarks, allows readers to gain confidence in using numerical methods for quantum dynamics. Given its educational value and clarity, I strongly recommend the publication of this article in SciPost.

I have a few minor remarks aimed at clarifying certain aspects and improving the manuscript’s overall readability and impact. They are listed below.

Requested changes

  1. It is unclear how spin is handled in the current implementation of MCTDH-X, for instance for distinguishing between spin-1/2 and spin-3/2 fermions. Given the importance of spin in problems like “chemical bonding” (which is mentioned with reference to natural bonding orbitals and Lewis structures), a comment on how spin could be set by the user and how spin-resolved properties can be analyzed would definitely improve clarity.

  2. While the energy expressions (Eq. 17 and 18) for the Harmonic Interaction Model are useful, a deeper explanation of how the many-body wavefunction is constructed from the one-particle solutions in the collective coordinates q_k's would enhance the reader’s understanding. Including a worked example for N = 3 particles (for both bosons and fermions) would be helpful.

  3. On page 14, it would be useful to mention if the software allows the user to define custom one- and two-body potentials. Additionally, the distinction between the keywords “Interaction_type” and “which_interaction” is not fully clear and could be elaborated upon.

  4. Along with Table 1, since the exact energy is known, it would be informative to plot the energy error against computation time on a log-linear scale. This would also apply to Fig. 5 and Fig. 6, where plotting \log_{10}(E - E_{\text{exact}}) vs imaginary time would better illustrate convergence behavior.

  5. For Figs. 7 and 8, it would be useful to compare the exact orbital occupancies if available. Also, a log-linear plot could be more insightful than the current linear-linear one. For Fig. 8, it should be made clear that the sum of the occupations adds up to one, not N, as might be expected from the general discussion in the introductory section.

  6. Several figure captions do not match the figure content. For instance, the caption for Fig. 10 refers to purple and green curves, but the figure itself shows blue and red curves. Furthermore, axis labels should be checked—e.g., in Fig. 13, “Double well” could be more appropriately labeled as “Potential” or “V”. Solid lines should be used consistently to improve readability (the different colors suffice to distinguish them)

  7. The discussion on symmetry-breaking and restoration raises the question of whether one can start with a \emph{minimal} grid (where symmetry is likely preserved) and progressively converge the system to a fully symmetric wavefunction. Could the authors include an illustrative calculation to clarify this? This is mentioned by the authors but a detailed example could be helpful.

  8. The density results for N = 5 fermions, showing five peaks, suggests formation of a “Wigner molecule” . Could the authors comment on this possibility and eventually extend their discussion of advanced tools, such as correlation functions, to include an analysis of the conditional probabilities that are readily available from the two-body correlation function?

  9. The notation for the one-body density matrix (Eq. 23) and the diagonal part of the two-body density matrix (Eq. 24) may be confusing, as they represent two very different functions of two spatial variables. Furthermore, the statement on p. 42 that “different points of \rho^{(1)} can be interpreted…as the probability amplitude of finding a particle at one point given that there is a particle at another point” should be reconsidered.

  10. The command on page 41 (”.\MCDTH_analysis_gcc”) seems incorrect. Furthermore, the rationale behind copying the executables into the working directory for each task is unclear. Could the authors explain this process more clearly?

Recommendation

Ask for minor revision

---

## Round 1 · Referee Report · Anonymous (Referee 2) · 2024-10-21

Strengths

  1. Explains how to use a useful code in a tutorial style. It will form a starting point for any student wishing to use the MCTDH-X program.
  2. Well written, with an overview of the theory and well chosen examples showing what can be calculated.
  3. The examples introduce the concepts of MCTDH-X in an easy-to-follow manner.

Weaknesses

  1. There a no obvious weaknesses in the paper. The code itself may be quite specialised and seems to require a good knowledge of linux.

Report

The lecture notes provide a tutorial for the MCTDH-X code. This is a code able to solve the Schrodinger equation (time-dependent or time-independent) for a system of particle (fermions or bosons) taking symmetry into account. This is an active area of research and it is still computationally challenging to compute a wavefunction for more than a few particles. The MCTDH-X method is one way of obtaining accurate results for systems of interest past the usual mean-field type approaches.

The tutorial takes the reader through how to set up the program and use it, with examples. Outputs as well as inputs are given to show what the program produces and how to analyse the results. Overall this will provide an excellent starting point to someone wishing to learn how to use the code. It is of use to the community and shows the present state-of-the-art of this method. I can thus recommend it for publication in the Lecture Notes of SciPost.

I have a couple of requests for making it clear upfront what the program is capable of and available documentation. It would also be interesting to know whether the data in the output files can be easily accessed for analysis, e.g .using python scripts. There are also a few typos.

Requested changes

In the preamble (or the introduction) it would be good to have a mention of available documentation. Where does one look for information going beyond the simple tutorial examples, and what does one expect to find?

In a similar way, it would be nice to have more details at the start on the potentials already available and whether / how it is possible to add custom potentials. or at least where to look if one wants to do this. Also, what is the format of the output data files? Is it possible to read them for user-defined analysis using, e.g. python?

Typos.
p4. top "one of such" -> "one such"
p5 below Eq.(3) "in TISE" -> "in the TISE"
Fig. 7 caption "inset show a" -> "inset shows a"
p22 "because of the for" -> "because for"
p32 "Note that while turbulent" -> "Note that, while turbulent"
"we will see also form" -> "we will see also for"
p37 "either site" -> "either side"

Recommendation

Publish (easily meets expectations and criteria for this Journal; among top 50%)

---

## Round 1 · Referee Report · Anonymous (Referee 3) · 2024-10-28

Strengths

1- Well-documented tutorial, ideal for graduate-level students. 2- MCTDHX theory summarized well at an adequate level for a tutorial.

Weaknesses

1- The examples are discussed too briefly. 2- The presentation of the results, in particularly the figures, could be improved.

Report

These lecture notes provide a brief introduction on the theory multi-configuration time-dependent Hartree method for fermions and bosons, together with a comprehensive tutorial for the MCTDH-X code that can be used to get a numerically-accurate wave functions for dynamical and ground state calculations. This work represents a nice complement to the documentation available on the project website (ultracold.org), where the theory is explained in more detail and the User's Guide give more information about keywords, convergence parameters, convergence behaviour. As such it represents a nice contribution that deserves publication.

Requested changes

  • Eq.(6): the hats are missing on the creation/annihilation operators. This is also the case in the text below Eq.(8). Since the wave function is denoted \hat{\Phi}, a change of notation (maybe even as simple as \hat{\phi}) would help the reader.

  • p6 below Eq 9: "transform a given wavefunction into a localized form". "localized" in what sense? It should be clarified, as it conveys different meaning in different communities.

  • p15 "We know discuss" -> "We now discuss"

  • p16-17: the command "plot "NOPR.out" u 1:7 w l lw 3" should appear directly below the text on p.16 and not after the figure and table.

  • many figures would be more readable if the y-axis was present in log scale. This is the case for energy convergence (fig.6), as well as for the density (figs. 9-16, 19-21).

  • in all figures where the density and the harmonic potential are represented (figs 11,12, 14-16, 19-22, 24-26), two separate y-axes are needed. This is simple using gnuplot (option "axis x1y2" when plotting).

  • fig.17: the choice of dashed green lines make it impossible to follow the dynamics. This should be changed.

  • p.38: The authors write: "This is a cat state, and can be fragile." This needs to be explained.

  • p.43: The last occurence of "mencoder" is not properly formatted.

Recommendation

Ask for minor revision

---

## Round 2 · Referee Report · Anonymous (Referee 1) · 2025-1-23

Strengths

  1. Clarity in presenting the basics of MCTDH-X theory: these lecture notes are structured to guide the reader through the Multiconfigurational Time-Dependent Hartree approach for both bosonic and fermionic systems. They clearly introduce the theoretical framework, emphasizing the contrast between MCTDH and mean-field theories like Gross-Pitaevskii or Hartree-Fock

  2. Effectiveness in guiding the reader through complicated many-body problems: the article provides comprehensive benchmarks and step-by-step workflows, including exact solutions for specific model problems (the Harmonic Interaction Model) . It not only introduces the time-independent and time-dependent cases but also illustrates how to handle complex systems using numerical methods, with a balance between practical and theoretical aspects

  3. Completeness: the coverage of both imaginary- (for ground state) and real-time dynamics, as well as advanced analysis features like correlation functions, is thorough. The authors further includes discussion of delicate topics like symmetry-breaking and restoration.

Weaknesses

None

Report

I have carefully reviewed the authors’ reply letter and the revised version of the manuscript. I appreciate the authors’ efforts to address all the concerns raised by the reviewers in the first round and to amend the manuscript accordingly. The final version is significantly improved, and the addition of the new graphs with a more appropriate log scale is particularly illuminating. I am confident that this manuscript will be a valuable contribution to the field, and I am pleased to recommend its publication

Requested changes

Minor issue remaining:

  • below eq. 9 in p 5, please reconsider the statement "In chemistry, these localized orbitals...". It is not obvious (and indeed it is not true) that the natural orbtial are localized. Localization ("of the eigenvectors") becomes possible when there is degeneracy in the eigenvalues, i.e. in the natural population, to within some fixed accuracy. It is this freedom that it is exploited in the localization process.

Recommendation

Publish (surpasses expectations and criteria for this Journal; among top 10%)

---

## Round 2 · Author Response

The reply to the referees is contained in the attached document. The document contains formulas that cannot be displayed as simple text.

---

## Round 2 · List of Changes

The reply to the referees is contained in the attached document.

---

## Round 3 · Author Response

Dear Editor,
We thank you for your editorial work and also the Referees for their valuable feedback.

We agree with the Referee's last comment and have agreed to remove the term "localized" in the sentence "In chemistry, these localized orbitals...".
We hope now our manuscript can be processed for publication.

Kind regards,
Paolo Molignini, Sunayana Dutta, Elke Fasshauer

---

## Round 3 · List of Changes

We have removed the term "localized" in the sentence "In chemistry, these localized orbitals...".

---

## Editorial Decision

published